# Tumors mimic the niche to inhibit neighboring stem cell differentiation

Yang Zhang[1], Yuejia Wang[1], Jinqiao Song[1], Lizhong Yan[1], Ziguang Wang[1], Dongze Song[1], Haojun Wang[1], Sining Yang[1], Liyuan Niu[1], Chang Sun[1], Hanning Zhang[1], Yudi Zhao[1], Shaowei Zhao[1,2]*

[1]Department of Genetics and Cell Biology, College of Life Sciences, Nankai University, Tianjin, China; [2]Nankai International Advanced Research Institute (SHENZHEN FUTIAN), Shenzhen, China

## eLife Assessment

This study provides **important** insights into how tumorous germline stem cells (GSCs) in the *Drosophila melanogaster* ovary can mimic niche function and suppress the differentiation of neighboring cells. The findings that GSC tumors can incorporate non-mutant cells and inhibit their differentiation are **compelling** and extend current understanding of stem cell-niche interactions. However, the evidence supporting the conclusion that GSC tumors produce BMP ligands to mediate this effect remains incomplete, due to concerns regarding the quality and interpretation of the HCR-FISH data.

*For correspondence:
swzhao@nankai.edu.cn

Competing interest: The authors declare that no competing interests exist.

**Abstract** Although it is well established that stem cells maintain tissue homeostasis while tumors disrupt it, the mechanisms by which tumors influence the development of nearby stem cells remain poorly understood. Using *Drosophila* ovaries as a model system, here we discovered that *bam* or *bgcn* mutant germline tumors inhibit the differentiation of neighboring wild-type germline stem cells (GSCs). Mechanistically, these tumor cells mimic the stem cell niche by secreting the bone morphogenetic protein (BMP) ligands Dpp and Gbb, but at reduced levels, resulting in moderate BMP signaling activation in adjacent GSCs. Such BMP signaling activation is sufficient to repress *bam* transcription, thereby blocking GSC differentiation. To our knowledge, this is the first example that tumors can functionally mimic a stem cell niche to inhibit the differentiation of neighboring wild-type stem cells. Similar regulatory paradigms may operate in mammalian tissues, including humans, during tumorigenesis.

## Introduction

The homeostasis of many tissues in our bodies is maintained by adult stem cells, but this balance can be disrupted by tumor cells. What occurs when tumorigenesis intersects with stem cell development? To address this question, a mosaic analysis model system is essential, where wild-type stem cells develop alongside tumor cells. *Drosophila* offers an exceptional model for such studies, as it allows for the efficient generation of mosaic clones through various established methods (*Germani et al., 2018*; *Pastor-Pareja and Xu, 2013*).

In *Drosophila* ovaries, germline stem cells (GSCs) play a crucial role in sustaining normal oogenesis and maintaining fertility (*Fuller and Spradling, 2007*; *Lin, 1997*). These GSCs reside in a specialized microenvironment known as the stem cell niche (hereafter referred to as niche) (*Xie and Spradling, 2000*). Typically, a GSC undergoes asymmetric division, generating two distinct daughter cells: one remains in the niche to self-renew as a GSC, while the other, called a cystoblast, exits the niche

and initiates differentiation. During the differentiation process, each cystoblast performs exactly four rounds of mitotic division with incomplete cytokinesis to produce 16 interconnected cystocytes, forming a germline cyst. In each germline cyst, only one germ cell is destined to become the oocyte, while the remaining 15 differentiate into nurse cells that support the development of the oocyte (*Figure 1A*; *Fuller and Spradling, 2007*; *Lin, 1997*). The principal niche signals are Bone morphogenetic protein (BMP) ligands, including Decapentaplegic (Dpp) and Glass bottom boat (Gbb), which are secreted by cap and terminal filament (TF) cells (*Chen and McKearin, 2003a*; *Li et al., 2016*; *Song et al., 2004*; *Xie and Spradling, 1998*; *Xie and Spradling, 2000*). These ligands activate BMP signaling in GSCs, leading to the transcriptional repression of *bag of marbles* (*bam*), a key gene that promotes differentiation. In contrast, BMP signaling is inactive in cystoblasts, allowing Bam to be expressed and to drive their differentiation (*Chen and McKearin, 2003a*; *Song et al., 2004*). Bam carries out this function in collaboration with its partner, Benign gonial cell neoplasm (Bgcn) (*Li et al., 2009*; *Ohlstein et al., 2000*).

GSCs mutant for *bam* or *bgcn* fail to differentiate and instead hyper-proliferate, forming a well-established *Drosophila* germline tumor model (*Lavoie et al., 1999*; *McKearin and Ohlstein, 1995*; *McKearin and Spradling, 1990*; *Niki and Mahowald, 2003*). Notably, these germline tumor cells competitively displace wild-type GSCs from the niche (*Jin et al., 2008*). The resulting displacement creates a microenvironment where wild-type GSCs are surrounded by tumor cells, providing an excellent model system to study stem cell behavior in tumor neighborhoods.

Here, we demonstrate that *bam* or *bgcn* mutant germline tumors inhibit the differentiation of neighboring wild-type GSCs by functionally mimicking the stem cell niche. This mechanism may be conserved in mammals, including humans, during tumorigenesis, where malignant cells could similarly disrupt normal stem cell development.

## Results

### Germline tumors inhibit the differentiation of neighboring wild-type GSCs

To generate *bam* or *bgcn* mutant germline clones, we employed either *nos>FLP/FRT* or *hs-FLP/FRT* systems that we previously established (*Zhang et al., 2023b*; *Zhao et al., 2018*). These two systems induce the expression of FLP recombinase either germline-specifically (*nos-GAL4-VP16/UASz-FLP*) or via heatshock (*hs-FLP*). The expressed FLP recombinase targets the *FRT* sites to mediate mitotic recombination on homologous chromosome arms, generating adjacent GFP-negative (*bam* or *bgcn* mutant) and GFP-positive (wild-type) germ cell populations (*Figure 1B*). Remarkably, we observed that many wild-type germ cells located outside the niche retained a GSC-like single-germ-cell (SGC) morphology (*Figure 1C and D*), even when encapsulated within egg chambers (*Figure 1—figure supplement 1*). Under normal conditions, GSCs that exit the niche differentiate into interconnected germline cysts, where germ cells are linked rather than remaining as individual, isolated cells (*Fuller and Spradling, 2007*; *Xie and Spradling, 2000*). To rule out the possibility that the SGC phenotype is an artifact caused by GFP expression, we repeated the experiments using RFP and arm-lacZ as alternative mosaic analysis markers. Consistent results were observed (*Figure 1E and F*), confirming that the phenotype is not attributable to GFP.

To further confirm that these SGCs exhibit GSC-like characteristics, we conducted anti-α-Spectrin immunofluorescent staining, a method that labels a germline-specific organelle known as the spectrosome in GSCs and cystoblasts, and the fusome in cystocytes. GSCs perform complete cell division, whereas cystocytes undergo incomplete cytokinesis, remaining interconnected through fusomes and ring canals. Consequently, spectrosomes appear as dot-like structures, while fusomes exhibit branched morphologies (*Figure 1G*; *Lin et al., 1994*). To accurately capture the three-dimensional (3D) architecture of spectrosomes and fusomes, we acquired z-stack images using confocal microscopy. Strikingly, these SGCs displayed dot-like spectrosomes, closely resembling those observed in wild-type GSCs and *bam* or *bgcn* mutant GSC-like tumor cells (*Figure 1H and I*). We also considered the possibility that SGCs might arise through the dedifferentiation of the cystocytes in germline cysts surrounded by germline tumors. If this were the case, such cystocytes would initially undergo complete cell division, leaving behind midbodies as markers of the late cytokinesis stage. When visualized by anti-α-Spectrin immunofluorescence, midbody appears as a central sphere that is slightly connected to two larger

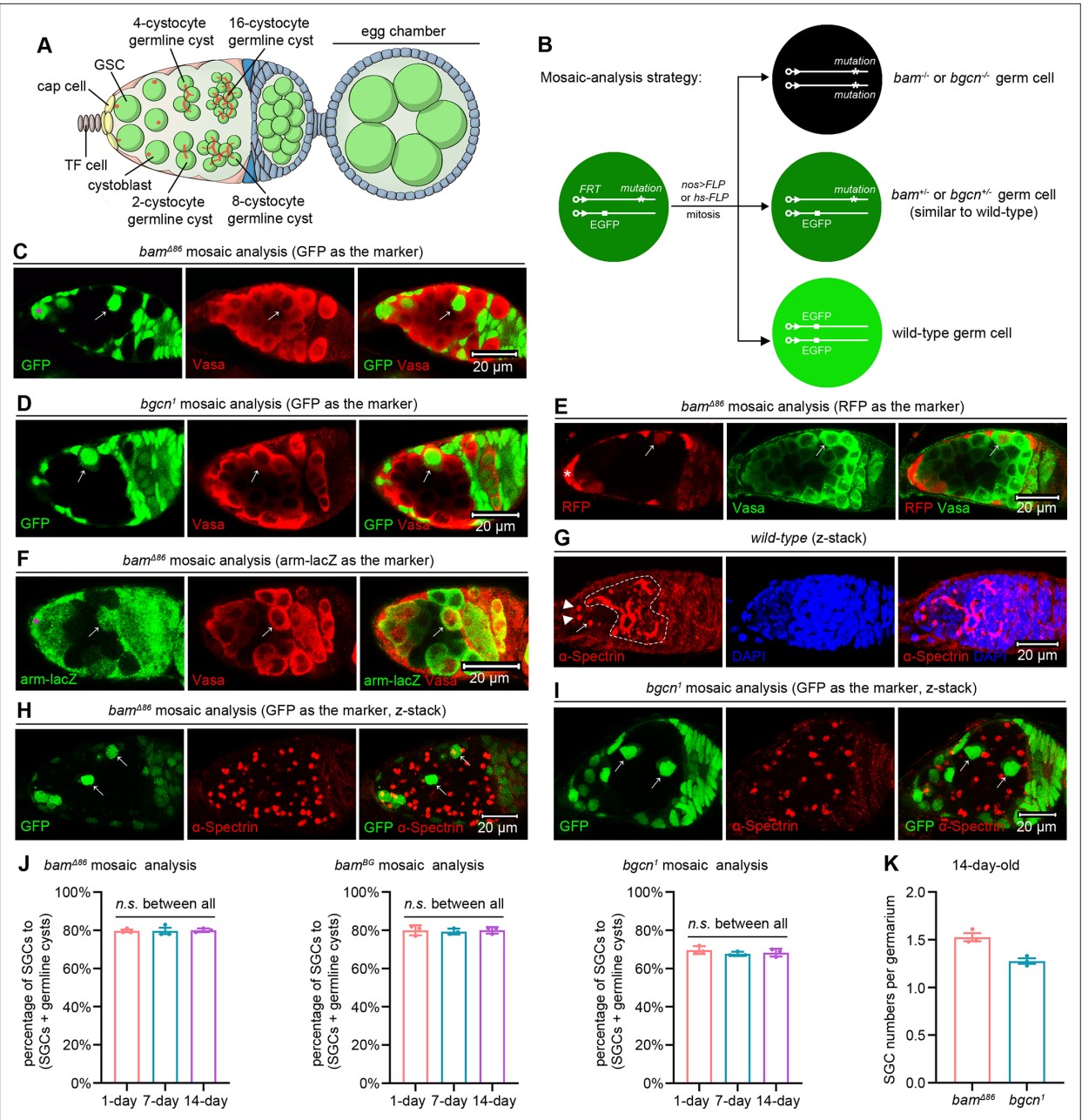

**Figure 1.** *bam* or *bgcn* mutant germline tumors inhibit the differentiation of neighboring wild-type GSCs. (**A**) Schematic cartoon for early oogenesis. The red dots and branches indicate spectrosomes and fusomes, respectively. TF cell: terminal filament cell; GSC: germline stem cell. (**B**) Mosaic analysis strategy. The FLP recombinase triggers mitotic recombination by targeting *FRT* sequences. The *nos>FLP* method restricts FLP expression to the germline, while the *hs-FLP* method enables heatshock-inducible FLP expression. (**C–F**) Representative samples. The asterisks mark cap cells, and the arrows indicate SGCs that have exited the niche and are surrounded by *bam* or *bgcn* mutant germline tumors. Vasa, a germ cell marker, should label all germ cells. However, due to poor tumor permeability, staining often fails to detect tumorous germ cells in the central region (see Vasa panels in D–F). (**G–I**) Representative samples (z-stack projections). In (**G**), the arrowheads and arrow, respectively, mark two GSCs and one cystoblast, all containing dot-like spectrosomes, while the dotted lines delineate cystocytes with branched fusomes. In (**H**) and (**I**), the arrows denote SGCs that also contain dot-like spectrosomes, akin to GSCs and the adjacent GSC-like tumor cells. (**J, K**) Quantification data. *bam^BG^* is a strong loss-of-function allele of *bam* (***Chen and McKearin, 2005***). For each experiment, three independent replicates were performed, and data represent mean ± SEM. In (**J**), over 100 SGCs and germline cysts were quantified per replicate, and statistical significance was determined by one-way ANOVA. *n.s.* ($P > 0.05$). In (**K**), over 100 germaria were quantified per replicate.

The online version of this article includes the following figure supplement(s) for figure 1:

**Figure supplement 1.** SGCs appear in egg chambers.

*Figure 1 continued on next page*

*Figure 1 continued*

**Figure supplement 2.** *bam* mutant germline clones enlarge as flies age.

**Figure supplement 3.** Comparison of SGC phenotypes induced by the *nos>FLP/FRT* and *hs-FLP/FRT* systems.

flanking structures, resembling a variant of nunchucks (*Mathieu et al., 2022*). Notably, in our analyses of over 50 germline cysts surrounded by *bam* mutant germline tumors, none contained midbodies, suggesting that dedifferentiation is unlikely to be the primary mechanism responsible for the SGC phenotype. Together, these findings indicate that *bam* or *bgcn* mutant germline tumors inhibit the differentiation of neighboring wild-type GSCs.

To quantify the SGC phenotype, which requires the presence of both germline tumors and out-of-niche wild-type germ cells, we analyzed germaria containing both. In 14-day-old fly ovaries, 70% of germaria (432/618) met this criterion. We calculated the percentage of SGCs relative to the

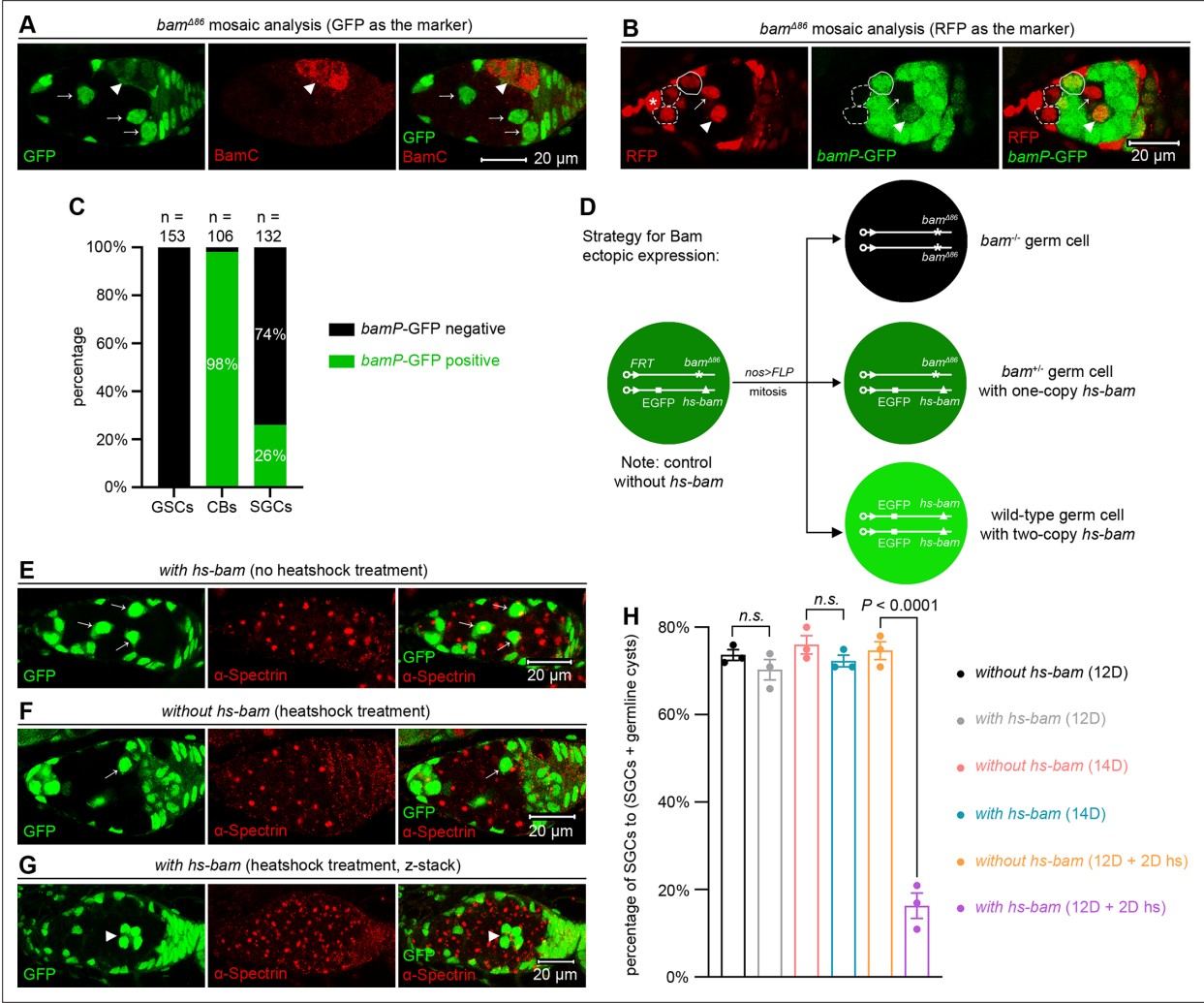

**Figure 2.** The inhibition of SGC differentiation depends on the lack of Bam expression. (**A**) Representative sample. The arrowhead marks a BamC-positive 4-cystocyte germline cyst, while the arrows indicate BamC-negative SGCs. (**B**) Representative sample. The asterisk denotes cap cells, and the dotted circles outline *bamP*-GFP-negative GSCs. The solid circle marks a *bamP*-GFP-positive cystoblast. The arrow and arrowhead point to *bamP*-GFP-negative and -positive SGCs, respectively. (**C**) Quantification data. 14-day-old flies were used for the analyses. CBs: cystoblasts. (**D**) Schematic of the experimental strategy for (**E–H**). In '*with hs-bam*' flies (**E, G**), wild-type germ cells (both *bam⁺/⁺* and *bam⁺/⁻*) carry the *hs-bam* transgene, while control '*without hs-bam*' flies (**F**) lack this element in their wild-type germ cells. (**E–G**) Representative samples. The arrows mark SGCs with dot-like spectrosomes, while the arrowhead indicates a 4-cystocyte germline cyst containing branched fusomes. (**H**) Quantification data. For each experiment, three independent replicates were performed, with over 100 SGCs and germline cysts quantified per replicate. Data represent mean ± SEM, and statistical significance was determined by t test. *n.s.* (*P* > 0.05).

total number of SGCs and germline cysts, considering the out-of-niche germ cells that are either fully enclosed by germline tumors (e.g. the right SGC in *Figure 1I* and the marked germline cyst in *Figure 2G*) or in contact with wild-type germ cells or somatic cells on only one side (e.g. the left SGC in *Figure 1I* and the germline cyst in the lower right corner of Figure 4C). Notably, the SGC phenotype was consistent across the 14-day period analyzed (*Figure 1J*). For either 1-, 7-, or 14-day time point, we measured the sizes of *bam* mutant germline clones in over 30 germaria containing these clones. To estimate 3D clone size, we counted cell numbers within the maximal 2D cross-sectional area of each clone. Clones were larger in 14-day-old flies than in either 1- or 7-day-old flies (*Figure 1—figure supplement 2*). Therefore, we selected the 14-day time point for all subsequent analyses to maximize experimental efficiency. In qualifying germaria, the average number of SGCs was approximately 1.5 (*Figure 1K*). For each biological replicate used to quantify the SGC phenotype, we counted more than 100 SGCs and germline cysts (>50 germaria analyzed). Furthermore, the SGC phenotypes induced by the *nos >FLP/FRT* and *hs-FLP/FRT* systems were indistinguishable (*Figure 1—figure supplement 3*). Given its simplicity and germline specificity, we primarily used the *nos>FLP/FRT* system in the following studies.

## The inhibition of differentiation in SGCs relies on the lack of Bam expression

Given that Bam is the key factor promoting GSC differentiation (*McKearin and Ohlstein, 1995*; *Ohlstein and McKearin, 1997*), we were very curious about the expression of Bam in SGCs. At first, we assessed Bam protein levels using immunofluorescent staining with an anti-BamC antibody (*McKearin and Ohlstein, 1995*). Strikingly, none of the SGCs examined (n > 100) were BamC-positive (*Figure 2A*). Then, we analyzed *bam* transcription levels using a *bamP*-GFP reporter (*Chen and McKearin, 2003b*). 100% of GSCs within the niche (n = 153) were GFP-negative, while 98% of cystoblasts (n = 106) were GFP-positive (*Figure 2B and C*), confirming that *bam* transcription is associated with the initiation of GSC differentiation (*McKearin and Ohlstein, 1995*). Notably, 74% of SGCs (n = 132) were GFP-negative (GSC-like), while the remaining 26% were GFP-positive (cystoblast-like) (*Figure 2B and C*). The cystoblast-like SGCs may have already initiated their differentiation program toward becoming cystocytes. Since *bam* transcription initiates in cystoblasts (*McKearin and Spradling, 1990*) but Bam proteins accumulate predominantly in cystocytes (*McKearin and Ohlstein, 1995*), the Bam protein levels in these cystoblast-like SGCs are likely below the detection threshold at this early stage.

Next, we asked whether ectopic expression of Bam can drive SGCs to differentiate. To address this, we established two experimental scenarios: one with the *hs-bam* element and one without as the control. In the *hs-bam* scenario (*with hs-bam*), GFP-positive germ cells are wild-type (carrying *hs-bam*), while GFP-negative cells are *bam* mutant (lacking *hs-bam*). In the control scenario (*without hs-bam*), GFP-positive cells are wild-type, and GFP-negative cells are *bam* mutant (*Figure 2D*, see genotypes in *Source data 1*). To induce ectopic Bam expression, 12-day-old female flies were subjected to heat-shock treatment, which involved heating at 37°C for 2 hr, twice daily with a 6 hr interval, and over 2 consecutive days. In the absence of heatshock treatment, the percentage of SGCs in ovaries of both genotypes showed no significant difference at either 12 or 14 days (*Figure 2E–H*), indicating that the *hs-bam* element alone, without heatshock, does not affect the phenotype. However, following heat-shock treatment, the percentage of SGCs in ovaries with *hs-bam* was markedly reduced compared to those without *hs-bam* (*Figure 2E–H*), suggesting that ectopic Bam expression can drive SGCs to differentiate. Collectively, these results support that the differentiation defects of SGCs are due to the lack of Bam expression.

## SGCs retain moderate BMP signaling activation

Within the niche, BMP signaling functions to repress *bam* transcription to inhibit GSC differentiation (*Chen and McKearin, 2003a*; *Song et al., 2004*). To investigate BMP signaling activation in SGCs, we employed immunofluorescent staining for pMad, a well-characterized marker of BMP signaling activity (*Kai and Spradling, 2003*). Surprisingly, we observed undetectable pMad levels in all SGCs examined (n > 100) (*Figure 3A, B*). To investigate this further, we examined the activity of *Dad*-lacZ, a highly sensitive BMP signaling reporter known to be activated not only in GSCs but also in cystoblasts (*Kai and Spradling, 2003*; *Song et al., 2004*). Notably, 73% of SGCs were lacZ-positive (n = 107), a proportion lower than that of GSCs within the niche, which showed 100% lacZ positivity (n = 122)

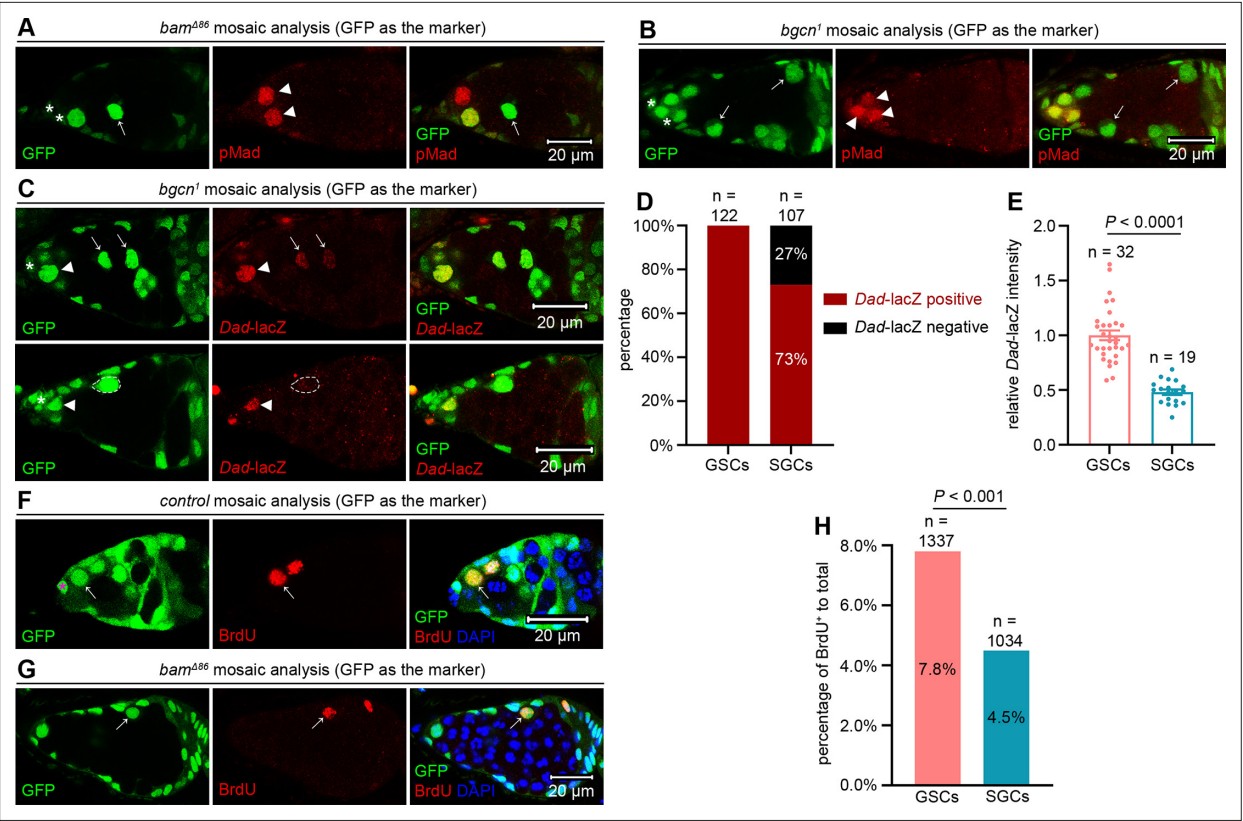

**Figure 3.** SGCs maintain lower BMP signaling levels than GSCs within the niche. (**A, B**) Representative samples. The asterisks mark cap cells, arrowheads indicate pMad-positive GSCs, and arrows point to pMad-negative SGCs. (**C**) Representative samples. The asterisks denote cap cells, arrowheads mark *Dad*-lacZ-positive GSCs, and arrows highlight *Dad*-lacZ-positive SGCs. The dotted cycles outline one *Dad*-lacZ-negative SGC. (**D, E**) Quantification data. 14-day-old flies were used for the analyses. In (**E**), data represent mean ± SEM, and statistical significance was determined by t test. (**F**) Representative sample. The asterisk marks a cap cell, while the arrows indicate a BrdU⁺ GSC within the niche. (**G**) Representative sample. The arrow indicates a BrdU⁺ SGC surrounded by germline tumors. (**H**) Quantification data. 14-day-old flies were used for the analyses. Statistical significance was determined by chi-squared test.

(*Figure 3C, D*). Furthermore, when comparing *Dad*-lacZ expression levels exclusively in lacZ-positive cells, we found that SGCs exhibited significantly lower expression levels than GSCs within the niche (*Figure 3C, E*). These findings indicate that BMP signaling is activated in SGCs but at lower levels than those in GSCs within the niche.

Beyond maintaining *Drosophila* female GSCs in the niche, BMP signaling also promotes their division (*Xie and Spradling, 1998*). Since the activation levels of BMP signaling in SGCs were lower than those in GSCs within the niche, we hypothesized that SGCs would exhibit slower proliferation rates than GSCs. To test this hypothesis, we performed BrdU incorporation assays. The results revealed that only 4.5% of SGCs were BrdU-positive (n = 1034), a significantly lower proportion than the 7.8% observed in GSCs within the niche (n = 1337) (*Figure 3F–H*). These findings further corroborate the reduced activation of BMP signaling in SGCs relative to GSCs.

## BMP signaling inhibits SGC differentiation

Then, we investigated whether BMP signaling functions to inhibit SGC differentiation. The BMP type II receptor Punt and the co-Smad Medea (Med) are essential for maintaining GSC stemness within the niche (*Xie and Spradling, 1998*). Therefore, we sought to determine whether they are also required to inhibit SGC differentiation. However, because distinguishing one versus two copies of GFP proved difficult in our germline mosaic assays, we established a genetic scenario, in which GFP⁺/⁺ RFP⁻/⁻ germ cells are *punt⁻/⁻* or *med⁻/⁻*; GFP⁺/⁻ RFP⁺/⁻ germ cells are *punt⁺/⁻ bam⁺/⁻* or *med ⁺/⁻ bam⁺/⁻* (similar to wild-type); and GFP⁻/⁻ RFP⁺/⁺ germ cells are *bam⁻/⁻*. In control experiments (with no *punt* or *med* mutation), GFP⁺/⁺ RFP⁻/⁻ germ cells are wild-type; GFP⁺/⁻ RFP⁺/⁻ germ cells are *bam⁺/⁻* (similar to wild-type);

and GFP$^{-/-}$ RFP$^{+/+}$ germ cells are *bam*$^{-/-}$ (*Figure 4A*, see genotypes in *Source data 1*). Strikingly, the proportion of *punt*$^{-/-}$ or *med*$^{-/-}$ SGCs relative to total SGCs was significantly lower than in controls (*Figure 4B–E*). Conversely, among *punt*$^{-/-}$ or *med*$^{-/-}$ germ cells meeting our established criteria for SGC phenotype quantification, germline cysts constituted a higher percentage compared to controls (*Figure 4F*). These results indicate that Punt and Med function to inhibit SGC differentiation.

Mothers against dpp (Mad) is the primary transcription factor of BMP signaling, and it is also essential for GSC maintenance in *Drosophila* ovaries (*Xie and Spradling, 1998*). Unlike *punt* and

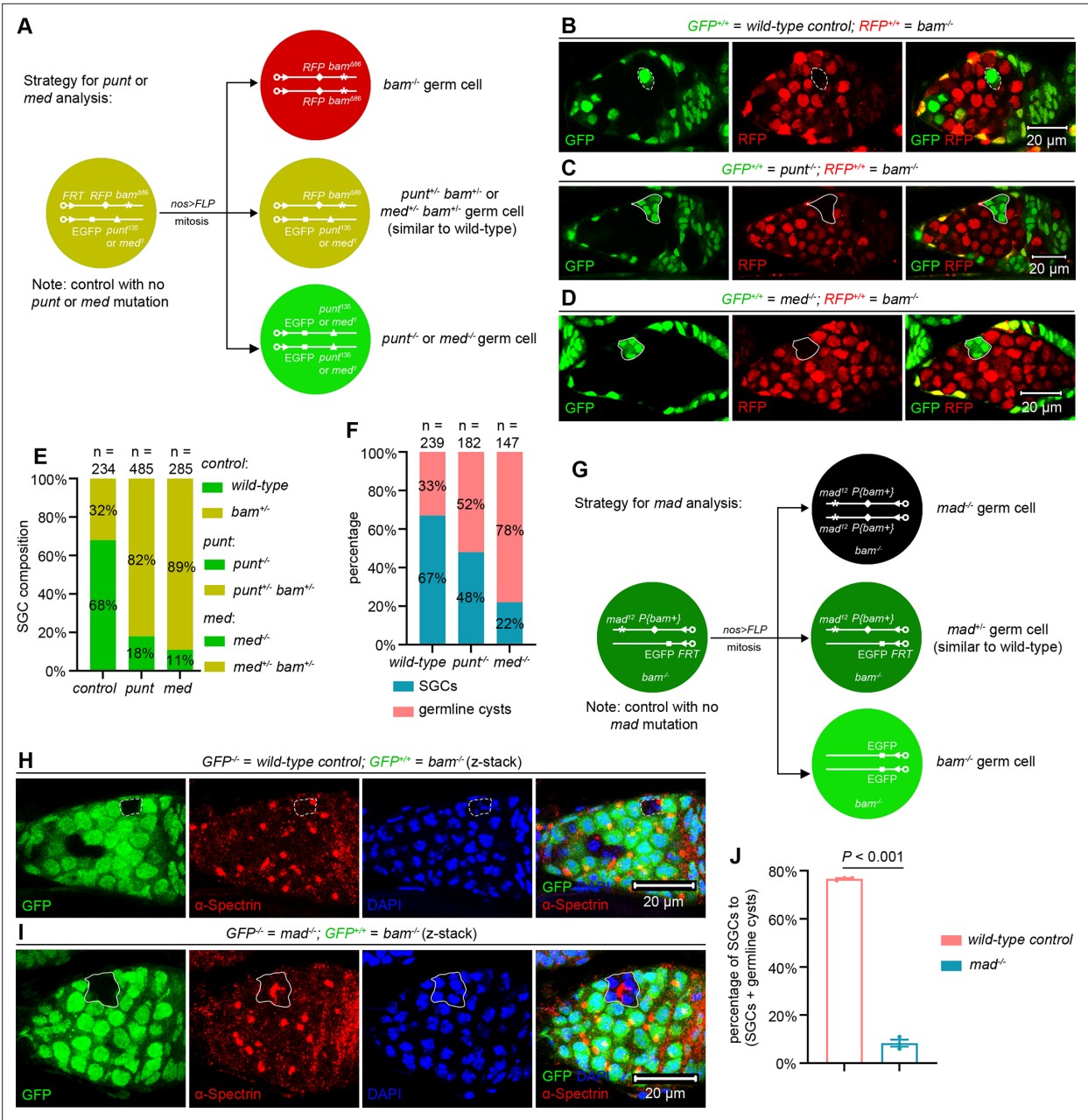

**Figure 4.** BMP signaling inhibits SGC differentiation. (**A**) Schematic of the experimental strategy for (**B–F**). Genotypes were unambiguously distinguished using a triple-color system (red, yellow, and green). (**B–D**) Representative samples. The dotted cycles mark an SGC, while the solid lines outline germline cysts containing differentiating cystocytes. (**E, F**) Quantification data. 14-day-old flies were used for the analyses. (**G**) Schematic of the experimental strategy for (**H–J**). (**H, I**) Representative samples. The dotted lines mark an SGC, while the solid lines outline a germline cyst containing differentiating cystocytes. (**J**) Quantification data. 14-day-old flies were used for the analyses. For each experiment, three independent replicates were performed, with over 100 SGCs and germline cysts quantified per replicate. Data represent mean ± SEM, and statistical significance was determined by t test.

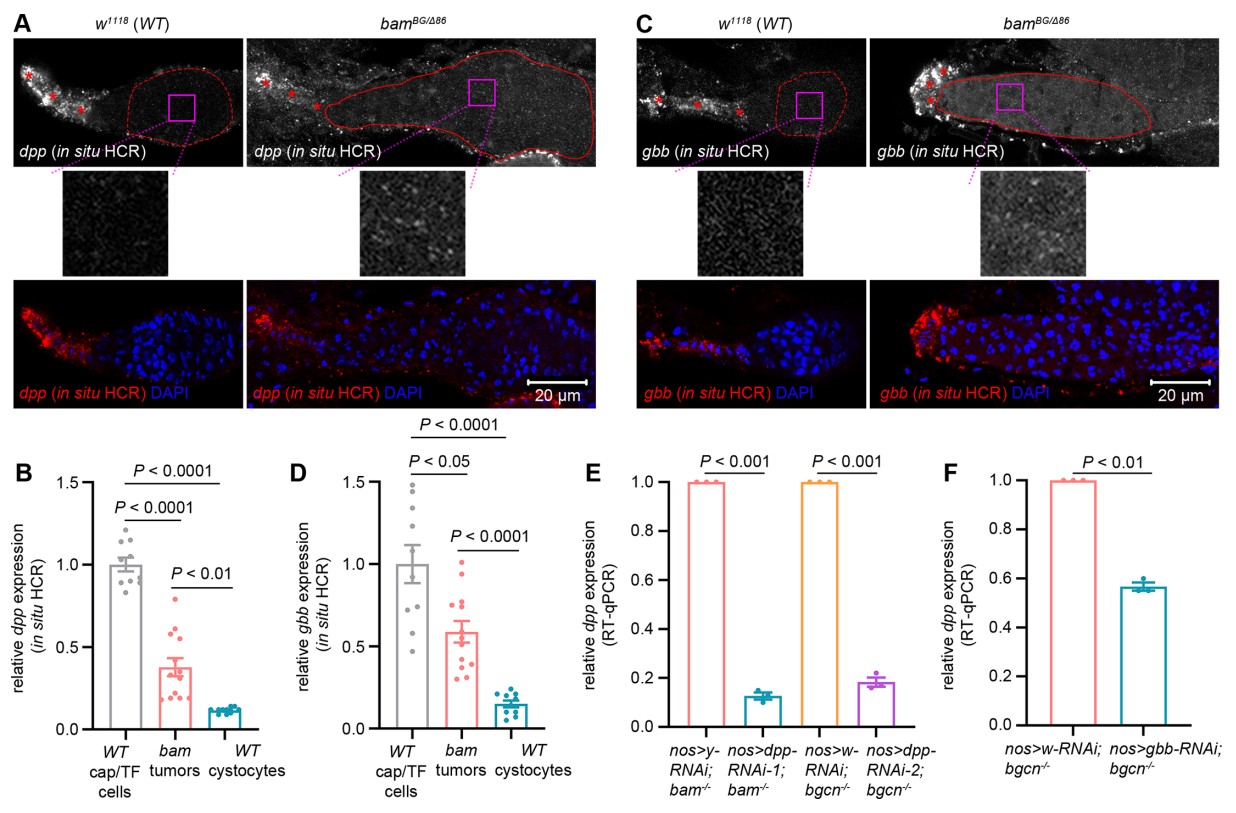

**Figure 5.** Germline tumors secrete Dpp and Gbb. (**A, C**) Representative samples. The asterisks denote cap/TF cells. The dotted lines highlight wild-type (WT) cystocytes, while the solid lines outline *bam* mutant germline tumor cells. The magenta box areas are enlarged below. (**B, D**) Quantification data for *in situ* HCR assays. 14-day-old flies were used for the analyses, and over 10 samples were quantified for each genotype. (**E, F**) Quantification data for RT-qPCR assays. 14-day-old flies were used for the analyses. For each experiment, three independent replicates were performed. Data represent mean ± SEM, and statistical significance in (**B, D**) was determined by one-way ANOVA and in (**E, F**) by t test.

*med*, which reside on the same chromosome arm (3R) as *bam*, *mad* is located on a separate chromosome arm (2L). To investigate whether Mad is required to inhibit SGC differentiation, we established a genetic scenario, in which GFP$^{-/-}$ germ cells are *mad*$^{-/-}$ and GFP$^{+/+}$ germ cells are *bam*$^{-/-}$. In control experiments (with no *mad* mutation), GFP$^{-/-}$ germ cells are wild-type and GFP$^{+/+}$ germ cells are *bam*$^{-/-}$ (*Figure 4G*, see genotypes in *Source data 1*). Notably, *mad* mutation significantly decreased the SGC proportion relative to controls (*Figure 4H–J*). These results suggest that, like Punt and Med, Mad also plays a crucial role in suppressing SGC differentiation. Together, these findings demonstrate that BMP signaling contributes to inhibiting SGC differentiation, despite at reduced activation levels.

## Germline tumors secrete Dpp and Gbb

The formation of a differentiation niche by escort cells is required for GSC differentiation and is known to be disrupted by *bam* mutant germline tumors (*Chen et al., 2022*; *Kirilly et al., 2011*). Although this niche disruption could contribute to the SGC phenotype, an unaddressed question is the source of the BMP ligands (Dpp and Gbb) that maintain BMP signaling activation within SGCs. Given that *dpp* expression has been detected in some *bam* mutant germline tumor cells from both *in vivo* and *in vitro* sources (*Niki et al., 2006*), we hypothesized that these tumor cells secrete BMP ligands to inhibit neighboring GSC differentiation. To assess the expression of *dpp* and *gbb*, we employed third-generation *in situ* hybridization chain reaction (HCR) (*Choi et al., 2018*). Successful detection was confirmed by prominent signal foci in cap and TF cells (*Figure 5A and C*). To enable quantitative comparison, all experiments and confocal imaging were performed under identical parameters. Signal intensity within *bam* mutant germline tumors and wild-type cystocytes was normalized to the signal in wild-type cap and TF cells. Strikingly, *bam* mutant germline tumor cells exhibited significantly elevated expression of both *dpp* and *gbb* compared to wild-type cystocytes (*Figure 5A–D*).

To more sensitively assess *dpp* and *gbb* expression, we performed real-time quantitative PCR (RT-qPCR) analyses in *bam* or *bgcn* mutant ovaries, comparing samples with and without germline-specific knockdown of *dpp* or *gbb*. Detection of reduced transcript levels in knockdown conditions would confirm active expression of these genes in the respective genetic backgrounds. Consistent with the essential roles of these two genes in fly viability, ubiquitous knockdown using *act-GAL4* with either *dpp-RNAi* or *gbb-RNAi* caused lethality, which also validated the efficacy of these RNAi lines. Notably, germline-specific knockdown of *dpp* or *gbb* significantly reduced their transcript levels compared to *yellow* (*y*) or *white* (*w*) knockdown controls (*Figure 5E and F*). Collectively, these findings demonstrate that *bam* or *bgcn* mutant germline tumors secrete the BMP ligands, albeit at lower levels than cap and TF cells.

## Dpp and Gbb secreted by germline tumors are required to inhibit SGC differentiation

Finally, we investigated whether Dpp and Gbb secreted by germline tumors are required to inhibit SGC differentiation. Using a previously established double-mutant mosaic analysis strategy for two genes on different chromosomes (*Zhang et al., 2024*; *Zhang et al., 2023b*), we generated *dpp bam* or *gbb bam* double-mutant germline clones using two *dpp* mutant alleles, $dpp^{d6}$, $dpp^{d12}$, and one *gbb* allele, $gbb^1$ (*Figure 6A and B*, see genotypes in *Source data 1*). Heterozygotes in any of these alleles did not affect GSC maintenance, germ cell differentiation, and female fly fertility (*Figure 6—figure supplement 1*). However, both *dpp bam* and *gbb bam* double-mutant germline tumor cells exhibited reduced proliferation rates compared to *bam* single-mutant controls (*Figure 6—figure supplement 2*), indicating that autocrine BMP signaling promotes *bam* mutant tumor growth. As mentioned earlier, our evaluation focused on germ cells that have exited the niche and are surrounded by germline tumors to quantify the SGC phenotype. Thus, it raises the question of whether the extent of tumor encirclement (i.e. being surrounded by more or fewer tumor cells) influences the phenotype. To investigate this, we compared the SGC phenotype in bigger and smaller *bam* mutant germline tumors. A total of 70 germaria containing *bam* mutant germline clones were analyzed using the same method described in *Figure 1—figure supplement 2B*. The 35 bigger and 35 smaller clones were categorized as 'bigger' and 'smaller' tumors, respectively. Strikingly, the SGC phenotype remained consistent between the two tumor groups (*Figure 6—figure supplement 3*), aligning with our earlier finding that this phenotype is stable over a 14-day period (*Figure 1J*), a timeframe sufficient for substantial germline tumor growth (*Figure 1—figure supplement 2*). These results suggest that direct contact between tumorous and wild-type germ cells, rather than tumor size, is the primary determinant of this phenotype.

The results above demonstrate that comparing the severity of the SGC phenotype is feasible between germ cells surrounded by smaller *dpp bam* or *gbb bam* double-mutant germline tumors and those surrounded by larger *bam* single-mutant germline tumors. Remarkably, both *dpp bam* and *gbb bam* double-mutant germline tumors enclosed fewer SGCs but more germline cysts than their *bam* single-mutant counterparts (*Figure 6C–I*). Thus, we concluded that the BMP ligands from directly-contacting germline tumor cells mediate the dominant inhibition of SGC differentiation. This amazingly parallels the mechanism observed in the normal stem cell niche, where only germ cells in direct contact with cap cells are maintained as GSCs (*Chen and McKearin, 2003a*; *Song et al., 2004*; *Xie and Spradling, 2000*).

## Discussion

Our study reveals that *bam* or *bgcn* mutant germline tumors in *Drosophila* ovaries secrete lower levels of BMP ligands Dpp and Gbb than cap and TF cells, resulting in moderate BMP signaling activation in adjacent wild-type GSCs (called SGCs in this study). Such BMP signaling activation is sufficient to repress *bam* transcription, thereby blocking SGC differentiation (see our working model in *Figure 7*). Strikingly, this mechanism closely recapitulates the normal niche signaling program mediated by cap and TF cells (*Chen and McKearin, 2003a*; *Song et al., 2004*; *Xie and Spradling, 1998*; *Xie and Spradling, 2000*). To our knowledge, this represents the first evidence that tumor cells can functionally mimic a stem cell niche to arrest neighboring wild-type stem cells in an undifferentiated state.

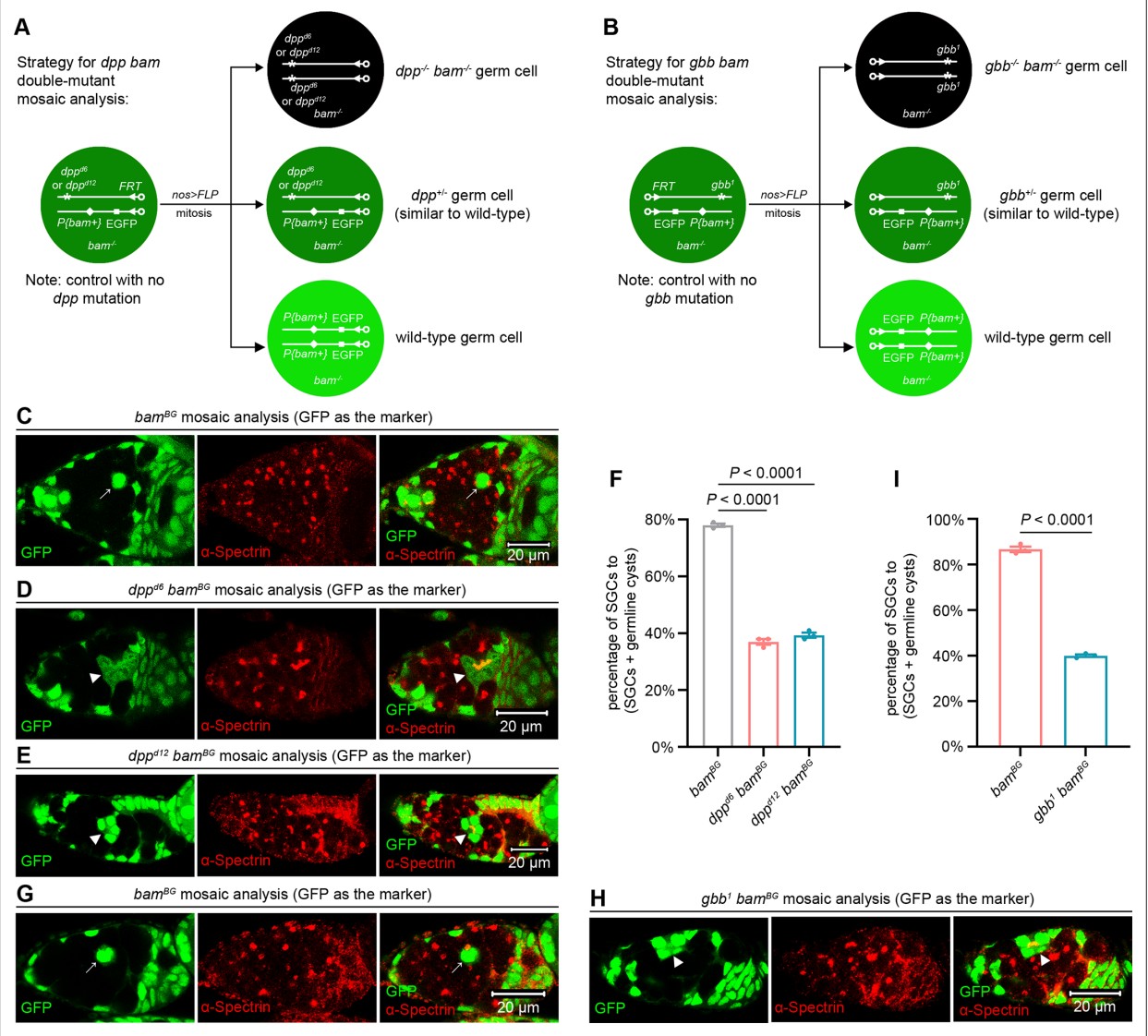

**Figure 6.** Dpp and Gbb secreted by germline tumors are required to inhibit SGC differentiation. (**A**) Schematic of the experimental strategy for (**C–F**). (**B**) Schematic of the experimental strategy for (**G–I**). (**C-E, G, H**) Representative samples. The arrows mark SGCs containing dot-like spectrosomes, while the arrowheads denote germline cysts with differentiating cystocytes that possess branched fusomes. (**F, I**) Quantification data for the SGC phenotype. 14-day-old flies were used for the analyses. For each experiment, three independent replicates were performed, with over 100 SGCs and germline cysts quantified per replicate. Data represent mean ± SEM. Statistical significance in (**F**) was determined by one-way ANOVA and in (**I**) by t test.

The online version of this article includes the following figure supplement(s) for figure 6:

**Figure supplement 1.** Monoallelic deletion of *dpp* or *gbb* does not affect GSC maintenance, germ cell differentiation, and female fly fertility.

**Figure supplement 2.** *dpp bam* or *gbb bam* double-mutant germline tumor cells divide more slowly than *bam* single-mutant ones.

**Figure supplement 3.** The SGC phenotype is unchanged irrespective of the number of surrounding germline tumors.

While *bam* or *bgcn* mutant germline tumors consist of GSC-like cells expected to resemble SGCs (*Lavoie et al., 1999*; *McKearin and Ohlstein, 1995*), we found key differences in BMP signaling. Out-of-niche *bgcn* mutant tumor cells showed significantly lower BMP activity than neighboring SGCs, as evidenced by reduced *Dad*-lacZ expression (*Figure 3C*). Consistent with this, most of the out-of-niche *bam* mutant tumor cells expressed *bamP*-GFP, a reporter suppressed by BMP signaling (*Chen and McKearin, 2003a*; *Song et al., 2004*), whereas only 26% of SGCs were *bamP*-GFP-positive (*Figure 2B and C*). These findings suggest that SGCs are more responsive to BMP signals secreted by germline tumors than the tumors themselves. Future studies are needed to elucidate the underlying mechanisms.

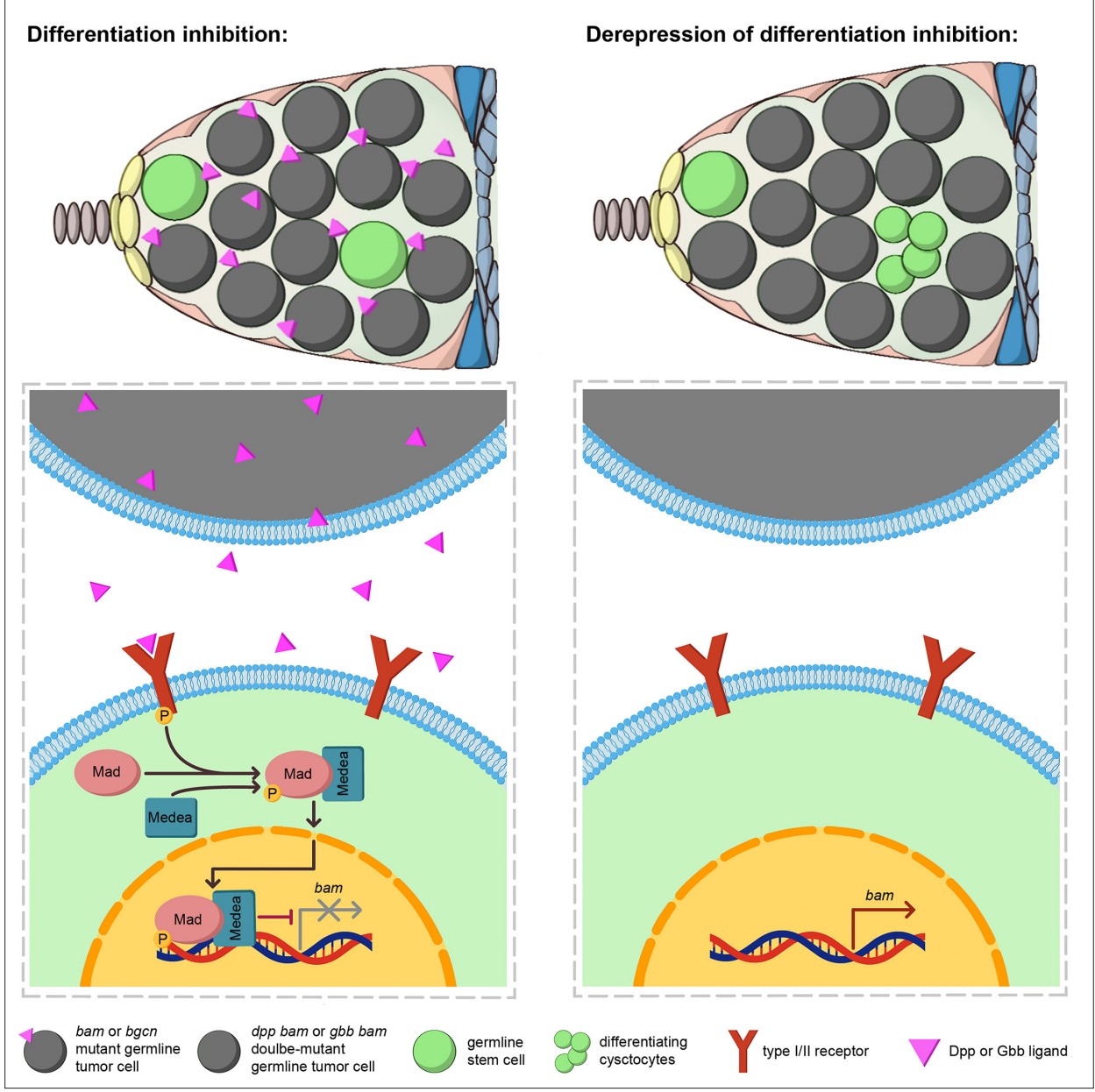

**Figure 7.** A working model. *bam* or *bgcn* mutant germline tumors secrete the BMP ligands Dpp and Gbb to activate BMP signaling in out-of-niche GSCs (called SGCs in this study) to inhibit their differentiation (left panel). In contrast, *dpp bam* and *gbb bam* double-mutant germline tumors exhibit a significant loss of this differentiation-inhibiting ability (right panel).

In the *Drosophila* ovarian germarium, the cell types that express *dpp* remain controversial. Two major approaches have been used to detect *dpp* transcription: *in situ* hybridization (ISH) and the *dpp*-lacZ reporter. An early, seminal study using ISH reported strong *dpp* transcription in developing follicle cells, with low levels in both cap and inner sheath cells (***Xie and Spradling, 2000***). In contrast, using either ISH or the *dpp*-lacZ reporter, some studies claimed that *dpp* is expressed exclusively in cap cells (***Luo et al., 2017***; ***Wang and Page-McCaw, 2018***). Several additional studies, also employing ISH or the *dpp*-lacZ reporter, detected strong *dpp* transcription in both cap and TF cells (***Li et al., 2016***; ***Liu et al., 2015***; ***Zhang et al., 2023a***), a pattern consistent with our *in situ* HCR data (***Figure 5A and C***). Notably, the cell type consistently identified across these studies is cap cell, the primary somatic cell comprising the stem cell niche (***Xie and Spradling, 2000***). These discrepancies in *dpp* expression

patterns may arise from differences in the ISH probes and *dpp* enhancer elements used, and further studies are clearly needed to resolve them.

One interesting finding is that *bam* or *bgcn* mutant germline tumors secrete lower levels of BMP ligands than cap and TF cells (*Figure 5A–D*). This aligns with earlier microarray data showing that purified *Drosophila* female GSCs express minimal Dpp and Gbb (*Kai et al., 2005*). However, our work reveals that such BMP levels in germline tumors are functionally critical to suppress SGC differentiation (*Figure 6*). Unlike normal GSCs, which receive unidirectional BMP ligands from cap cells (*Chen and McKearin, 2003a*; *Li et al., 2016*; *Song et al., 2004*; *Xie and Spradling, 2000*), SGCs are often fully surrounded by *bam* or *bgcn* mutant germline tumors. This spatial advantage likely enables tumors to inhibit SGC differentiation efficiently without matching the high BMP output of cap and TF cells. Moreover, since BMP signaling is known to both inhibit normal GSC differentiation and promote their proliferation (*Xie and Spradling, 1998*), it should similarly stimulate SGC expansion, which is detrimental for *bam* or *bgcn* mutant germline tumors. We propose that these tumor cells finely regulate BMP secretion to balance these opposing demands: maintaining differentiation blockade of SGCs while avoiding stimulation of their excessive proliferation.

A well-established principle in oncology is that tumor aggressiveness correlates with poor differentiation, with less-differentiated tumors exhibiting enhanced transformative capacity and metastatic potential (*Jögi et al., 2012*; *Lytle et al., 2018*). In *Drosophila* ovaries, *bam* or *bgcn* mutant germline tumors consist of GSC-like cells that may resemble these poorly differentiated human tumors (*Lavoie et al., 1999*; *McKearin and Ohlstein, 1995*). This similarity raises the possibility that stem cell-like human tumors may similarly inhibit the differentiation of adjacent wild-type stem cells. By blocking differentiation, such tumors could deplete terminally differentiated cell populations, potentially exacerbating patient mortality. This mechanism may contribute to the heightened lethality of poorly differentiated tumors. Further investigation is needed to test this hypothesis.

The differentiation of a single GSC into a 16-cell germline cyst, comprising 15 polyploid nurse cells and 1 developing oocyte, represents a substantial metabolic investment (*Fuller and Spradling, 2007*; *Lin, 1997*). We propose that *bam* or *bgcn* mutant germline tumors block this process to divert nutrients toward their own uncontrolled growth. This phenomenon could have broad implications, as many human tissues and organs (intestine, muscle, skin, blood system, male germline, etc.) similarly depend on adult stem cells for homeostasis (*Blanpain and Fuchs, 2006*; *Gehart and Clevers, 2019*; *Sousa-Victor et al., 2022*; *Spradling et al., 2011*; *Wilkinson et al., 2020*). Notably, these stem cell-dependent tissues and organs are frequent sites of tumorigenesis, raising the possibility that human cancers may similarly impair neighboring stem cell differentiation to optimize nutrient allocation for malignant growth. A key limitation of our study is that the evidence is derived solely from *Drosophila* germline. Future work should explore whether similar regulatory paradigms operate in mammalian tissues during tumorigenesis.

## Materials and methods

### Fly husbandry
Flies were raised at 25°C on standard cornmeal/molasses/agar media.

### Transgenic flies
*hs-bam* on chromosome 3R: The coding sequence of the *bam* gene, amplified from the cDNA clone, was cloned into the *BglII-XbaI* sites of the *pCaSpeR-hs* vector, while the *attB* sequence was inserted into the *XhoI* site. The resulting *attB-pCaSpeR-hs-bam* plasmid was then microinjected into the *attP154* (Chromosome 3R, *97D2*) fly strain to generate site-specific transgenic flies.

### Heatshock method to induce germline clones
To ensure developmental synchrony and maintain low-density growth, eggs within 8 hr of laying were collected for heatshock treatment. The animals (late-Larva 3/early-Pupa stage) were subjected to twice-daily heatshocks at 37°C (2 hr per session, with a 6 hr interval between the two sessions) for 6 consecutive days.

### Fertility test
For each genotype, three independent crosses were performed. Each cross vial contained two females and four *w^1118* (wild-type) males, all aged 3 days old. The crosses were transferred to fresh vials every

2 days, with five replicate vials quantified per genotype. After all adult flies eclosed, offspring production was assessed by counting the number of empty pupae on the vial walls.

## BrdU labeling

Ovaries were dissected in Schneider's insect medium (SIM) and incubated in freshly prepared BrdU solution (100 µg/mL in SIM) for 5 hr at 25°C. After washing with PBS for 30 min, samples were fixed in 4% paraformaldehyde (in PBS) for 3 hr, followed by another PBS wash for 30 min. Samples were then treated with RQ1 DNase reaction solution (Promega, Madison, WI, USA) for 1 hr, washed with PBST (0.3% Triton X-100 in PBS) for 30 min, and incubated overnight at 4°C with mouse anti-BrdU antibody. Following a PBST wash for 1 hr, ovaries were incubated with goat anti-mouse 546 and DAPI (0.1 µg/mL) in PBST for 3 hr, washed again in PBST for 1 hr, and mounted in autoclaved 70% glycerol.

## Immunofluorescent staining, image collection, and data processing

Ovaries were dissected in PBS, fixed in 4% paraformaldehyde (in PBS) for 3 hr, washed with PBST for 30 min, and then incubated overnight at 4°C with primary antibodies. The rabbit anti-pMad antibody was a gift from Ed Laufer, and the rabbit anti-Vasa antibody was a gift from Zhaohui Wang (*Chen et al., 2014*). After washing with PBST for 1 hr, samples were incubated with Alexa Fluor-conjugated secondary antibodies and 0.1 µg/mL DAPI (in PBST) for 3 hr, followed by a final PBST wash for 1 hr. Ovaries were mounted in autoclaved 70% glycerol and imaged using a Zeiss LSM 710 confocal microscope (Carl Zeiss AG, Baden-Württemberg, Germany). Images were processed with ZEN 3.0 SR imaging software (Carl Zeiss) and Adobe Photoshop 2025. The quantification data were processed by GraphPad Prism, ImageJ, or Microsoft Excel.

## *In situ* HCR assay

Ovaries dissected from 14-day-old female flies were processed according to the following protocol.

1. Fixation: Ovaries were fixed in 4% paraformaldehyde (in PBS) for 3 hr at room temperature (RT) or overnight at 4°C.
2. Hybridization: Following fixation, samples were washed three times for 5 min each in PBST, dehydrated in methanol for 5 min, rehydrated through a methanol:PBST gradient series (3:1, 1:1, and 1:3), followed by another three 5 min PBST washes, treated with Proteinase K (10 µg/mL) for 5 min, washed again three times for 5 min each in PBST, pre-hybridized in preheated hybridization buffer (50% formamide, 5× SSC, 9 mM citric acid [pH 6.0], 0.1% Tween 20, 50 µg/mL heparin, 1× Denhardt's solution, 10% dextran sulfate) for 30 min at 37°C, and then incubated with *in situ* HCR probes (0.1 µM in hybridization buffer) overnight at 37°C. For the detection of *dpp* and *gbb*, a pool of 20 *in situ* HCR probes targeting each mRNA was employed. The probe sequences were provided in the Key resources table.
3. Signal amplification: The next day, samples were washed four times for 15 min each at 37°C with preheated probe wash buffer (50% formamide, 5× SSC, 9 mM citric acid, 0.1% Tween 20, 50 µg/mL heparin), followed by three 10 min washes in 5× SSCT (5× SSC, 0.1% Tween 20) at RT. After pre-hybridization in amplification buffer (5× SSC, 0.1% Tween 20, 10% sodium sulfate) for 10 min at RT, an amplification reaction was performed using heat-denatured hairpin nucleic acids (30 nM for each in amplification buffer) overnight in the dark at RT. The hairpin sequences were provided in the Key resources table.
4. Washing and mounting: Samples were washed three times for 10 min each in 5× SSCT, followed by three 10 min washes in PBST, and then mounted in autoclaved 70% glycerol for imaging.

## Quantification of the *in situ* HCR assay

The 2D cross-sectional germarium images containing cap/TF regions were captured using confocal microscopy with identical parameters. For each wild-type (*w^1118^*) germarium image, the cap/TF and cystocyte regions were outlined separately; for each *bam* mutant (*bam^BG/Δ86^*) germarium, the entire germline region was outlined. Mean fluorescence intensities from these regions were measured using ImageJ to assess the expression levels of *dpp* and *gbb*. For both wild-type cystocytes and *bam* mutant germline tumor cells, these expression levels were normalized to the average levels measured in wild-type cap/TF cells. Given that nearly no background signal was observed (compare germline with empty regions in wild-type germaria in *Figure 5A and C*), background subtraction was not applied. Over 10 germaria were quantified for each genotype.

## Real-time quantitative PCR

Ovaries from 14-day-old flies were dissected, and total RNA was extracted using the RNeasy Micro Kit. Equal amounts of RNA were reverse-transcribed into cDNA using the HiFiScript cDNA Synthesis Kit. RT-qPCR was performed on a CFX Connect Real-Time PCR System (Bio-Rad) with ChamQ SYBR qPCR Master Mix. The PCR protocol consisted of an initial denaturation at 95°C for 30 min, followed by 40 cycles of 95°C for 10 s and 60°C for 30 s. Relative gene expression was calculated using the $2^{-\Delta\Delta CT}$ method (*Livak and Schmittgen, 2001*). The primers used, which were previously described (*Huang et al., 2017*), were listed in the Key resources table.

## Acknowledgements

We thank Eric Baehrecke, Michael Buszczak, Zheng Guo, Ed Laufer, Ruth Lehmann, Weiwei Liu, Rongwen Xi, Ting Xie, Zhaohui Wang, Guojie Zhang, BDGP, BDSC, DSHB, and THFC for providing antibodies, plasmids, fly strains, and technical assistance. This study was supported by grants 32270841 and 32070871 from the National Natural Science Foundation of China (NSFC) to Shaowei Zhao.

## Additional information

### Funding

| Funder | Grant reference number | Author |
| --- | --- | --- |
| National Natural Science Foundation of China | 32270841 | Shaowei Zhao |
| National Natural Science Foundation of China | 32070871 | Shaowei Zhao |

The funders had no role in study design, data collection and interpretation, or the decision to submit the work for publication.

### Author contributions

Yang Zhang, Data curation, Software, Formal analysis, Validation, Investigation, Methodology, Writing – review and editing; Yuejia Wang, Jinqiao Song, Lizhong Yan, Ziguang Wang, Dongze Song, Haojun Wang, Sining Yang, Liyuan Niu, Chang Sun, Hanning Zhang, Yudi Zhao, Data curation, Formal analysis, Validation, Investigation, Methodology, Writing – review and editing; Shaowei Zhao, Conceptualization, Resources, Data curation, Software, Formal analysis, Supervision, Funding acquisition, Validation, Investigation, Visualization, Methodology, Writing – original draft, Project administration, Writing – review and editing

### Author ORCIDs

Shaowei Zhao ⓘ https://orcid.org/0000-0002-4544-7215

Reviewer #1 (Public review): https://doi.org/10.7554/eLife.108910.4.sa1
Reviewer #2 (Public review): https://doi.org/10.7554/eLife.108910.4.sa2
Reviewer #3 (Public review): https://doi.org/10.7554/eLife.108910.4.sa3
Author response https://doi.org/10.7554/eLife.108910.4.sa4

## Additional files

### Supplementary files

MDAR checklist

Source data 1. All genotypes.

Source data 2. Raw quantification data.

**Data availability**

All genotypes are described in *Source data 1*, and the raw quantification data are included in *Source data 2*. Fly strains and plasmids are available upon request.

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

# Appendix 1

**Appendix 1—key resources table**

| Reagent type (species) or resource | Designation | Source or reference | Identifiers | Additional information |
|---|---|---|---|---|
| Genetic reagent (*Drosophila melanogaster*) | *act-GAL4* | FlyBase Reference Report: Tepass, 2016.9.10, P{Act-GAL4.U} insertion. | | |
| Genetic reagent (*D. melanogaster*) | *bam^BG* (strong loss-of-function allele) | **Chen and McKearin, 2005** | | |
| Genetic reagent (*D. melanogaster*) | *bam^Δ86* (null allele) | Bloomington *Drosophila* Stock Center (BDSC) | 5427 | |
| Genetic reagent (*D. melanogaster*) | *bamP-GFP* | **Chen and McKearin, 2003b** | | |
| Genetic reagent (*D. melanogaster*) | *bgcn^1* (null allele) | BDSC | 6054 | |
| Genetic reagent (*D. melanogaster*) | *bgcn^MI06696* (strong loss-of-function allele) | BDSC | 40815 | |
| Genetic reagent (*D. melanogaster*) | *Canton-S* | BDSC | 64349 | |
| Genetic reagent (*D. melanogaster*) | *Dad-lacZ* | **Kai and Spradling, 2003** | | |
| Genetic reagent (*D. melanogaster*) | *dpp^d6* (hypomorphic allele) | BDSC | 2062 | |
| Genetic reagent (*D. melanogaster*) | *dpp^d12* (hypomorphic allele) | BDSC | 2070 | |
| Genetic reagent (*D. melanogaster*) | *EGFP FRT40A* | BDSC | 5629 | |
| Genetic reagent (*D. melanogaster*) | *FRT40A* | BDSC | 1816 | |
| Genetic reagent (*D. melanogaster*) | *FRT42D* | BDSC | 1802 | |
| Genetic reagent (*D. melanogaster*) | *FRT42D EGFP* | BDSC | 5626 | |
| Genetic reagent (*D. melanogaster*) | *FRT82B* | BDSC | 86313 | |
| Genetic reagent (*D. melanogaster*) | *FRT82B arm-lacZ* | BDSC | 7369 | |
| Genetic reagent (*D. melanogaster*) | *FRT82B EGFP* | BDSC | 32655 | |
| Genetic reagent (*D. melanogaster*) | *FRT82B RFP* | BDSC | 30555 | |
| Genetic reagent (*D. melanogaster*) | *gbb^1* (null allele) | BDSC | 98344 | |
| Genetic reagent (*D. melanogaster*) | *hs-bam on chromosome 3R* | This paper | | Construction information described in the Materials and methods section |
| Genetic reagent (*D. melanogaster*) | *mad^12* (null allele) | BDSC | 51301 | |

*Appendix 1 Continued on next page*

*Appendix 1 Continued*

| Reagent type (species) or resource | Designation | Source or reference | Identifiers | Additional information |
|---|---|---|---|---|
| Genetic reagent (*D. melanogaster*) | *med¹* (null allele) | BDSC | 9033 | |
| Genetic reagent (*D. melanogaster*) | *nos-GAL4-VP16* | BDSC | 4937 | |
| Genetic reagent (*D. melanogaster*) | *P{bam+}* | **Zhang et al., 2023b** | | |
| Genetic reagent (*D. melanogaster*) | *punt¹³⁵* (strong loss-of-function allele) | BDSC | 3100 | |
| Genetic reagent (*D. melanogaster*) | *UASp-dpp-RNAi-1* | TsingHua Fly Center (THFC) | TH201500984.S | |
| Genetic reagent (*D. melanogaster*) | *UASp-dpp-RNAi-2* | THFC | THU5880 | |
| Genetic reagent (*D. melanogaster*) | *UASp-FLP* | **Zhang et al., 2023b** | | |
| Genetic reagent (*D. melanogaster*) | *UASp-gbb-RNAi* | THFC | THU1480 | |
| Genetic reagent (*D. melanogaster*) | *UASp-GFP* | **Zhang et al., 2024** | | |
| Genetic reagent (*D. melanogaster*) | *UASp-yellow-RNAi* | THFC | TH03150.N | |
| Genetic reagent (*D. melanogaster*) | *UASz-FLP* | **Zhang et al., 2023b** | | |
| Genetic reagent (*D. melanogaster*) | *w¹¹¹⁸* | BDSC | 3605 | |
| Antibody | Anti-α-Spectrin (Mouse monoclonal) | Developmental Studies Hybridoma Bank (DSHB) | RRID:AB_528473 | IF (1:100) |
| Antibody | Anti-BamC (Mouse monoclonal) | DSHB | RRID:AB_10570327 | IF (1:5) |
| Antibody | Anti-β-Gal | DSHB | RRID:AB_528101 | IF (1:200) |
| Antibody | Anti-BrdU (Mouse monoclonal) | Sigma | B5002 | IF (1:400) |
| Antibody | Anti-pMad (Rabbit polyclonal) | **Zhao et al., 2018** | | A gift from Ed Laufer, IF (1:500) |
| Antibody | Anti-Vasa (Rabbit polyclonal) | **Chen et al., 2014** | | A gift from Zhaohui Wang, IF (1:2000) |
| Antibody | Alexa Fluor 546 goat anti-mouse | Invitrogen | Cat# A-11030 | IF (1:1000) |
| Antibody | Alexa Fluor 546 goat anti-rabbit | Invitrogen | Cat# A11035 | IF (1:1000) |
| Antibody | Goat anti-rabbit 488 | Apexbio | K1206 | IF (1:1000) |
| Recombinant DNA reagent | *bam* cDNA clone | Berkeley *Drosophila* Genome Project | FI05606 | |
| Recombinant DNA reagent | *pCaSpeR-hs* | *Drosophila* Genomics Resource Center | RRID:DGRC_1215 | |

*Appendix 1 Continued on next page*

*Appendix 1 Continued*

| Reagent type (species) or resource | Designation | Source or reference | Identifiers | Additional information |
|---|---|---|---|---|
| Recombinant DNA reagent | *attB-pCaSpeR-hs-bam* | This paper | | Construction information described in the Materials and methods section |
| Sequence-based reagent | The hairpin sequence for *in situ* HCR | This paper | B1H1-594 | CGTAAAGGAAGACTCTTCCCGTTTGCTG CCCTCCTCG CATTCTTTCTTGAGGAGGGCAGCAAACG GGAAGAG |
| Sequence-based reagent | The hairpin sequence for *in situ* HCR | This paper | B1H2-594 | GAGGAGGGCAGCAAACGGGAAGAGTCTT CCTTTACG CTCTTCCCGTTTGCTGCCCTCCTCAAGA AAGAATGC |
| Sequence-based reagent | *dpp* probe for *in situ* HCR | This paper | | GAGGAGGGCAGCAAACGGaaAGAGCATG GCCACGCTGTCCAGTTG |
| Sequence-based reagent | *dpp* probe for *in situ* HCR | This paper | | GCACCACCGTACTTTGGTCGTTGAGtaGAAGA GTCTTCCTTTACG |
| Sequence-based reagent | *dpp* probe for *in situ* HCR | This paper | | GAGGAGGGCAGCAAACGGaaTCGTAGCC CAGAGGCGCCACAATCC |
| Sequence-based reagent | *dpp* probe for *in situ* HCR | This paper | | GGGCACTTCCCGTGGCAGTAATATGtaGAAGA GTCTTCCTTTACG |
| Sequence-based reagent | *dpp* probe for *in situ* HCR | This paper | | GAGGAGGGCAGCAAACGGaaTCTCGGCT GCCGCTTGTTCCGGCCG |
| Sequence-based reagent | *dpp* probe for *in situ* HCR | This paper | | GTCGTGGTTCTTGCGCCTCGTAGGCtaG AAGAGTCTTCCTTTACG |
| Sequence-based reagent | *dpp* probe for *in situ* HCR | This paper | | GAGGAGGGCAGCAAACGGaaGCTGCTTG TGCTGCCACCGCTCGTG |
| Sequence-based reagent | *dpp* probe for *in situ* HCR | This paper | | CGTCGTCCGTGTAGGTGAACAGGAGtaG AAGAGTCTTCCTTTACG |
| Sequence-based reagent | *dpp* probe for *in situ* HCR | This paper | | GAGGAGGGCAGCAAACGGaaGGCCGGCT GGACATCGAGGCTCACC |
| Sequence-based reagent | *dpp* probe for *in situ* HCR | This paper | | CTGCGGACTCGCCAGCCACCGGTCCtaG AAGAGTCTTCCTTTACG |
| Sequence-based reagent | *dpp* probe for *in situ* HCR | This paper | | GAGGAGGGCAGCAAACGGaaCACCTGGT AGCGCGTCCGATTCGCC |
| Sequence-based reagent | *dpp* probe for *in situ* HCR | This paper | | CCCGACGCGCGTGATGTCGTAGACAtaG AAGAGTCTTCCTTTACG |
| Sequence-based reagent | *dpp* probe for *in situ* HCR | This paper | | GAGGAGGGCAGCAAACGGaaTGCTCTTC ACGTCGAAGTGCAGCCG |
| Sequence-based reagent | *dpp* probe for *in situ* HCR | This paper | | CCGCCTTCAGCTTCTCGTCGGCGGGtaG AAGAGTCTTCCTTTACG |
| Sequence-based reagent | *dpp* probe for *in situ* HCR | This paper | | GAGGAGGGCAGCAAACGGaaCCCATGAT CTCGGCGTAGAGCTTCT |
| Sequence-based reagent | *dpp* probe for *in situ* HCR | This paper | | GGGATGTTGACCGAGTCGAGCTCGTtaG AAGAGTCTTCCTTTACG |
| Sequence-based reagent | *dpp* probe for *in situ* HCR | This paper | | GAGGAGGGCAGCAAACGGaaAGCGCCTC CTTGCTGTAGGTGGACG |
| Sequence-based reagent | *dpp* probe for *in situ* HCR | This paper | | GGGTCTGGCTTCAGCTTGTCCTTGAtaGAAGA GTCTTCCTTTACG |
| Sequence-based reagent | *dpp* probe for *in situ* HCR | This paper | | GAGGAGGGCAGCAAACGGaaCACGAAGA TTGATTCAATCGACGAG |

*Appendix 1 Continued on next page*

*Appendix 1 Continued*

| Reagent type (species) or resource | Designation | Source or reference | Identifiers | Additional information |
|---|---|---|---|---|
| Sequence-based reagent | *dpp* probe for *in situ* HCR | This paper | | GCGGTCGAGCACCAGCGTCGGCTCCtaG AAGAGTCTTCCTTTACG |
| Sequence-based reagent | *gbb* probe for *in situ* HCR | This paper | | GAGGAGGGCAGCAAACGGaaGGTGGTAC AGAACGGGTAGTGCTCC |
| Sequence-based reagent | *gbb* probe for *in situ* HCR | This paper | | TTTTCAGGTTCACATTCTCGTCGTTtaGAAGA GTCTTCCTTTACG |
| Sequence-based reagent | *gbb* probe for *in situ* HCR | This paper | | GAGGAGGGCAGCAAACGGaaGCGTTCAT GTGCGCATTGAGCGGGA |
| Sequence-based reagent | *gbb* probe for *in situ* HCR | This paper | | AGGGTCTGGACGATCGCATGGTTCGtaG AAGAGTCTTCCTTTACG |
| Sequence-based reagent | *gbb* probe for *in situ* HCR | This paper | | GAGGAGGGCAGCAAACGGaaGTACAGGG TCTGCATCTGGCAGCTG |
| Sequence-based reagent | *gbb* probe for *in situ* HCR | This paper | | ATGCCAGCCCAGATCCTTGAAGTCTtaGAAGA GTCTTCCTTTACG |
| Sequence-based reagent | *gbb* probe for *in situ* HCR | This paper | | GAGGAGGGCAGCAAACGGaaTGCTCCTG TGGTGGCTGCTGTGGGC |
| Sequence-based reagent | *gbb* probe for *in situ* HCR | This paper | | GCTTGCGTGGATGGCTGGCGCTTCGtaG AAGAGTCTTCCTTTACG |
| Sequence-based reagent | *gbb* probe for *in situ* HCR | This paper | | GAGGAGGGCAGCAAACGGaaATGTCGTC CAGCTTCACCTCGCGGT |
| Sequence-based reagent | *gbb* probe for *in situ* HCR | This paper | | TCGTCCACCTTGCGGTGGATCAGTCtaG AAGAGTCTTCCTTTACG |
| Sequence-based reagent | *gbb* probe for *in situ* HCR | This paper | | GAGGAGGGCAGCAAACGGaaGTTGAGCT CCAACCAGCCCACGTAG |
| Sequence-based reagent | *gbb* probe for *in situ* HCR | This paper | | CAGCCACTCGTGCAGGCCCTCGGTCtaG AAGAGTCTTCCTTTACG |
| Sequence-based reagent | *gbb* probe for *in situ* HCR | This paper | | GAGGAGGGCAGCAAACGGaaACTCCCTG TTGGCGGTCAGCCACTT |
| Sequence-based reagent | *gbb* probe for *in situ* HCR | This paper | | TGCCAATGGCGTATACCGTGATGGTtaGAAGA GTCTTCCTTTACG |
| Sequence-based reagent | *gbb* probe for *in situ* HCR | This paper | | GAGGAGGGCAGCAAACGGaaCGACGGCC GTGCTCGTGACGCAGTT |
| Sequence-based reagent | *gbb* probe for *in situ* HCR | This paper | | GGCACGTTGGAGACGTCGAACCACAtaG AAGAGTCTTCCTTTACG |
| Sequence-based reagent | *gbb* probe for *in situ* HCR | This paper | | GAGGAGGGCAGCAAACGGaaGTTCTTCT GCTGCTCGCCCTCATCC |
| Sequence-based reagent | *gbb* probe for *in situ* HCR | This paper | | GGCCCGCTTGTCCAGGTCGGTGATGtaG AAGAGTCTTCCTTTACG |
| Sequence-based reagent | *gbb* probe for *in situ* HCR | This paper | | GAGGAGGGCAGCAAACGGaaTGATGCGG TGGTAGACGTCCAGCAG |
| Sequence-based reagent | *gbb* probe for *in situ* HCR | This paper | | CCTGATCGCTGAGACCCTCCTCCGCtaG AAGAGTCTTCCTTTACG |
| Sequence-based reagent | RT-qPCR primer | *Huang et al., 2017* | *dpp* primer-1 | TACCACGCCATCCACTCAAC |
| Sequence-based reagent | RT-qPCR primer | *Huang et al., 2017* | *dpp* primer-2 | GCTCGTTACTCGATACGGCT |
| Sequence-based reagent | RT-qPCR primer | *Huang et al., 2017* | *gbb* primer-1 | CTGGATCATCGCACCAGAGG |

*Appendix 1 Continued on next page*

*Appendix 1 Continued*

| Reagent type (species) or resource | Designation | Source or reference | Identifiers | Additional information |
|---|---|---|---|---|
| Sequence-based reagent | RT-qPCR primer | *Huang et al., 2017* | *gbb* primer-2 | GTCTGGACGATCGCATGGTT |
| Sequence-based reagent | RT-qPCR primer | *Huang et al., 2017* | *rp49* (internal control) primer-1 | CACCGGATTCAAGAAGTTCC |
| Sequence-based reagent | RT-qPCR primer | *Huang et al., 2017* | *rp49* (internal control) primer-2 | GACAATCTCCTTGCGCTTCT |
| Commercial assay or kit | ChamQ SYBR qPCR Master Mix | Vazyme | Q311 | |
| Commercial assay or kit | HiFiScript cDNA Synthesis Kit | CWBIO | CW2569M | |
| Commercial assay or kit | RNeasy Micro Kit | QIAGEN | 74004 | |
| Software, algorithm | Adobe Photoshop 2025 | San Jose, CA, USA | RRID:SCR_014199 | |
| Software, algorithm | ImageJ | NIH | RRID:SCR_003070 | |
| Software, algorithm | GraphPad Prism | GraphPad Software, Inc | RRID:SCR_002798 | |

