## [Editor Report · eLife Assessment]

This study provides **important** insights into how tumorous germline stem cells (GSCs) in the *Drosophila melanogaster* ovary can mimic niche function and suppress the differentiation of neighboring cells. The findings that GSC tumors can incorporate non-mutant cells and inhibit their differentiation are **compelling** and extend current understanding of stem cell-niche interactions. However, the evidence supporting the conclusion that GSC tumors produce BMP ligands to mediate this effect remains incomplete, due to concerns regarding the quality and interpretation of the HCR-FISH data.

---

## [Referee Report · Reviewer #1 (Public review)]

Summary:

This preprint from Shaowei Zhao and colleagues presents results that suggest tumorous germline stem cells (GSCs) in the Drosophila ovary mimic the ovarian stem cell niche and inhibit the differentiation of neighboring non-mutant GSC-like cells. The authors use FRT-mediated clonal analysis driven by a germline-specific gene (nos-Gal4, UASp-flp) to induce GSC-like cells mutant for bam or bam's co-factor bgcn. Bam-mutant or bgcn-mutant germ cells produce tumors in the stem cell compartment (the germarium) of the ovary (Fig. 1). These tumors contain non-mutant cells - termed SGC for single germ cells. 75% of SGCs do not exhibit signs of differentiation (as assessed by bamP-GFP) (Fig. 2). The authors demonstrate that block in differentiation in SGC is a result of suppression of bam expression (Fig. 2). They present data suggesting that in 73% of SGCs BMP signaling is low (assessed by dad-lacZ) (Fig. 3) and proliferation is less in SGCs vs GSCs. They present genetic evidence that mutations in BMP pathway receptors and transcription factors suppress some of the non-autonomous effects exhibited by SGCs within bam-mutant tumors (Fig. 4). They show data that bam-mutant cells secrete Dpp, but this data is not compelling (see below) (Fig. 5). They provide genetic data that loss of BMP ligands (dpp and gbb) suppresses the appearance of SGCs in bam-mutant tumors (Fig. 6). Taken together, their data support a model in which bam-mutant GSC-like cells produce BMPs that act on non-mutant cells (i.e., SGCs) to prevent their differentiation, similar to what in seen in the ovarian stem cell niche. This preprint from Shaowei Zhao and colleagues presents results that suggest tumorous germline stem cells (GSCs) in the Drosophila ovary mimic the ovarian stem cell niche and inhibit the differentiation of neighboring non-mutant GSC-like cells. The authors use FRT-mediated clonal analysis driven by a germline-specific gene (nos-Gal4, UASp-flp) to induce GSC-like cells mutant for bam or bam's co-factor bgcn. Bam-mutant or bgcn-mutant germ cells produce tumors in the stem cell compartment (the germarium) of the ovary (Fig. 1). These tumors contain non-mutant cells - termed SGC for single germ cells. 75% of SGCs do not exhibit signs of differentiation (as assessed by bamP-GFP) (Fig. 2). The authors demonstrate that block in differentiation in SGC is a result of suppression of bam expression (Fig. 2). They present data suggesting that in 73% of SGCs BMP signaling is low (assessed by dad-lacZ) (Fig. 3) and proliferation is less in SGCs vs GSCs. They present genetic evidence that mutations in BMP pathway receptors and transcription factors suppress some of the non-autonomous effects exhibited by SGCs within bam-mutant tumors (Fig. 4). They show data that bam-mutant cells secrete Dpp, but this data is not compelling (see below) (Fig. 5). They provide genetic data that loss of BMP ligands (dpp and gbb) suppresses the appearance of SGCs in bam-mutant tumors (Fig. 6). Taken together, their data support a model in which bam-mutant GSC-like cells produce BMPs that act on non-mutant cells (i.e., SGCs) to prevent their differentiation, similar to what in seen in the ovarian stem cell niche.

Strengths:

(1) Use of an excellent and established model for tumorous cells in a stem cell microenvironment

(2) Powerful genetics allow them to test various factors in the tumorous vs non-tumorous cells

(3) Appropriate use of quantification and statistics

Weaknesses:

(1) What is the frequency of SGCs in nos>flp; bam-mutant tumors? For example, are they seen in every germarium, or in some germaria, etc or in a few germaria.

This concern was addressed in the rebuttal. The line number is 106, not line 103.

(2) Does the breakdown in clonality vary when they induce hs-flp clones in adults as opposed to in larvae/pupae?

This concern was addressed in the rebuttal. However, these statements are no on lines 331-335 but instead starting on line 339. Please be accurate about the line numbers cited in the rebuttal. They need to match the line numbers in the revised manuscript.

(3) Approximately 20-25% of SGCs are bam+, dad-LacZ+. Firstly, how do the authors explain this? Secondly, of the 70-75% of SGCs that have no/low BMP signaling, the authors should perform additional characterization using markers that are expressed in GSCs (i.e., Sex lethal and nanos).

The authors did not perform additional staining for GSC-enriched protein like Sex lethal and nanos.

(4) All experiments except Fig. 1I (where a single germarium with no quantification) were performed with nos-Gal4, UASp-flp. Have the authors performed any of the phenotypic characterizations (i.e., figures other than figure 1) with hs-flp?

In the rebuttal, the authors stated that they used nos>flp for all figures except for Fig. 1I. It would be more convincing for them to prove in Fig. 1 than there is not phenoytpic difference between the two methods and then switch to the nos>FLP method for the rest of the paper.

(5) Does the number of SGCs change with the age of the female? The experiments were all performed in 14-day old adult females. What happens when they look at young female (like 2-day old). I assume that the nos>flp is working in larval and pupal stages and so the phenotype should be present in young females. Why did the authors choose this later age? For example, is the phenotype more robust in older females? or do you see more SGCs at later time points?

The authors did not supply any data to prove that the clones were larger in 14-day-old flies than in younger flies. Additionally, the age of "younger" flies was not specified. Therefore, the authors did not satisfactorily answer my concern.

(6) Can the authors distinguish one copy of GFP versus 2 copies of GFP in germ cells of the ovary? This is not possible in the Drosophila testis. I ask because this could impact on the clonal analyses diagrammed in Fig. 4A and 4G and in 6A and B. Additionally, in most of the figures, the GFP is saturated so it is not possible to discern one vs two copies of GFP.

In the rebuttal, the authors stated that they cannot differential one vs two copies of GFP. They used other clone labeling methods in Fig. 4 and 6. I think that the authors should make a statement in the manuscript that they cannot distinguish one vs two copies of GFP for the record.

(7) More evidence is needed to support the claim of elevated Dpp levels in bam or bgcn mutant tumors. The current results with dpp-lacZ enhancer trap in Fig 5A,B are not convincing. First, why is the dpp-lacZ so much brighter in the mosaic analysis (A) than in the no-clone analysis (B); it is expected that the level of dpp-lacZ in cap cells should be invariant between ovaries and yet LacZ is very faint in Fig. 5B. I think that if the settings in A matched those in B, the apparent expression of dpp-lacZ in the tumor would be much lower and likely not statistically significantly. Second, they should use RNA in situ hybridization with a sensitive technique like hybridization chain reactions (HCR) - an approach that has worked well in numerous Drosophila tissues including the ovary.

The HCR FISH in Fig.5 of the revised manuscript needs an explanation for how the mRNA puncta were quantified. Currently, there is no information in the methods. What is meant but relative dpp levels. I think that the authors should report in and unbiased manner "number" of dpp or gbb puncta in TFs. For the germaria, I think that they should report the number of puncta of dpp or gbb divide by the total area in square pixels counted. Additionally, the background fluorescence is noticeably much higher in bamBG/delta86 germaria, which would (falsely) increase the relative intensity of dpp and gbb in bam mutants. Although, I commend the authors for performing HCR FISH, these data are still not convincing to me.

(8) In Fig 6, the authors report results obtained with the bamBG allele. Do they obtain similar data with another bam allele (i.e., bamdelta86)?

The authors did not try any experiments with the bamdelta86 allele, despite this allele being molecularly defined, where the bamBG allele is not defined.

Comments on second revision:

The authors have adequately addressed several points. However, there is still no information in the material and methods for how they measured and quantified the HCR-FISH probe signal. They have the same size region that they use for each genotype, but they do not control for the number of nuclei in each square. I would also be helpful if they provided a different image for the gbb probe stained in the mutant background. It is the only panel that does not have other germaria in very close proximity. I am still not fully convinced of the HCR data, esp for gbb.

---

## [Referee Report · Reviewer #2 (Public review)]

In the current version, Zhang et al. have made substantial improvements to the manuscript. It is now easier to read, and the data are more solid compared with the previous version, supporting their conclusion that tumor GSCs secrete stemness factors (BMPs and Dpp) to suppress the differentiation of neighboring wild-type GSCs. This study should benefit a broad readership across developmental biology, germ cell biology, stem cell biology, and cancer biology.

Comments on revision:

If the exact number of germaria was not recorded (as described), an approximate number can be provided in the Materials and Methods; for example, stating that more than 10 germaria were analyzed per biological replicate.

---

## [Referee Report · Reviewer #3 (Public review)]

Zhang et al. investigated how germline tumors influence the development of neighboring wild-type (WT) germline stem cells (GSC) in the Drosophila ovary. They report that germline tumors generated by differentiation-arrested mutations (bam and bgcn) inhibit the differentiation of neighboring WT GSCs by arresting them in an undifferentiated state, resulting from reduced expression of the differentiation-promoting factor Bam. They find that these tumor cells produce low levels of the niche-associated signaling molecules Dpp and Gbb, which suppress bam expression and consequently inhibit the differentiation of neighboring WT GSCs non-cell-autonomously. Based on these findings, the authors propose that germline tumors mimic the niche to suppress the differentiation of the neighboring wild-type germline stem cells.

Strengths:

The study uses a well-established in vivo model to addresses an important biological question concerning the interaction between germline tumor cells and wild-type (WT) germline stem cells in the Drosophila ovary. If the findings are substantiated, this study could provide valuable insights that are applicable to other stem cell systems.

Weaknesses:

The authors have addressed some of my concerns in the revised submission. However, the data presented do not allow the authors to distinguish whether the failed differentiation of WT stem cells/germline cells results from "arrested differentiation due to the loss of the differentiation niche" or from "direct inhibition by tumor-derived expression of niche-associated molecules Dpp and Gbb". The critical supporting data, HCR in situ results, are not sufficiently convincing.

---

## [Author Response]

The following is the authors’ response to the previous reviews

**eLife Assessment**
This study presents results supporting a model that tumorous germline stem cells (GSCs) in the Drosophila ovary mimic the stem cell niche and inhibit the differentiation of neighboring cells. The valuable findings show that GSC tumors often contain non-mutant cells whose differentiation is suppressed by the GSC tumorous cells. However, the evidence showing that the GSC tumors produce BMP ligands to suppress differentiation of non-mutant cells is incomplete due to concerns about the new HCR data.

Thanks for this assessment. All concerns raised by the reviewers regarding the HCR data and others are followed by our responses below.

**Public Reviews:**

**Reviewer #1 (Public review):**
Summary:This preprint from Shaowei Zhao and colleagues presents results that suggest tumorous germline stem cells (GSCs) in the Drosophila ovary mimic the ovarian stem cell niche and inhibit the differentiation of neighboring non-mutant GSC-like cells. The authors use FRT-mediated clonal analysis driven by a germline-specific gene (nos-Gal4, UASp-flp) to induce GSC-like cells mutant for bam or bam's co-factor bgcn. Bam-mutant or bgcn-mutant germ cells produce tumors in the stem cell compartment (the germarium) of the ovary (Fig. 1). These tumors contain non-mutant cells - termed SGC for single-germ cells. 75% of SGCs do not exhibit signs of differentiation (as assessed by bamP-GFP) (Fig. 2). The authors demonstrate that block in differentiation in SGC is a result of suppression of bam expression (Fig. 2). They present data suggesting that in 73% of SGCs BMP signaling is low (assessed by dad-lacZ) (Fig. 3) and proliferation is less in SGCs vs GSCs. They present genetic evidence that mutations in BMP pathway receptors and transcription factors suppress some of the non-autonomous effects exhibited by SGCs within bam-mutant tumors (Fig. 4). They show data that bam-mutant cells secrete Dpp, but this data is not compelling (see below) (Fig. 5). They provide genetic data that loss of BMP ligands (dpp and gbb) suppresses the appearance of SGCs in bam-mutant tumors (Fig. 6). Taken together, their data support a model in which bam-mutant GSC-like cells produce BMPs that act on non-mutant cells (i.e., SGCs) to prevent their differentiation, similar to what in seen in the ovarian stem cell niche. This preprint from Shaowei Zhao and colleagues presents results that suggest tumorous germline stem cells (GSCs) in the Drosophila ovary mimic the ovarian stem cell niche and inhibit the differentiation of neighboring non-mutant GSC-like cells. The authors use FRT-mediated clonal analysis driven by a germline-specific gene (nos-Gal4, UASp-flp) to induce GSC-like cells mutant for bam or bam's co-factor bgcn. Bam-mutant or bgcn-mutant germ cells produce tumors in the stem cell compartment (the germarium) of the ovary (Fig. 1). These tumors contain non-mutant cells - termed SGC for single-germ cells. 75% of SGCs do not exhibit signs of differentiation (as assessed by bamP-GFP) (Fig. 2). The authors demonstrate that block in differentiation in SGC is a result of suppression of bam expression (Fig. 2). They present data suggesting that in 73% of SGCs BMP signaling is low (assessed by dad-lacZ) (Fig. 3) and proliferation is less in SGCs vs GSCs. They present genetic evidence that mutations in BMP pathway receptors and transcription factors suppress some of the non-autonomous effects exhibited by SGCs within bam-mutant tumors (Fig. 4). They show data that bam-mutant cells secrete Dpp, but this data is not compelling (see below) (Fig. 5). They provide genetic data that loss of BMP ligands (dpp and gbb) suppresses the appearance of SGCs in bam-mutant tumors (Fig. 6). Taken together, their data support a model in which bam-mutant GSC-like cells produce BMPs that act on non-mutant cells (i.e., SGCs) to prevent their differentiation, similar to what in seen in the ovarian stem cell niche.Strengths:(1) Use of an excellent and established model for tumorous cells in a stem cell microenvironment(2) Powerful genetics allow them to test various factors in the tumorous vs non-tumorous cells(3) Appropriate use of quantification and statistics

Thank you for your valuable comments, and we greatly appreciate them.

Weaknesses:(1) What is the frequency of SGCs in nos>flp; bam-mutant tumors? For example, are they seen in every germarium, or in some germaria, etc or in a few germaria.This concern was addressed in the rebuttal. The line number is 106, not line 103.(2) Does the breakdown in clonality vary when they induce hs-flp clones in adults as opposed to in larvae/pupae?This concern was addressed in the rebuttal. However, these statements are no on lines 331-335 but instead starting on line 339. Please be accurate about the line numbers cited in the rebuttal. They need to match the line numbers in the revised manuscript.

We have rechecked the line numbers and confirmed that the mismatch arose from the Word-to-PDF conversion process on the *eLife* website. As this issue has recurred and reviewers’ file-format preferences are unknown to us, we have added a clarifying note at the beginning of each response letter: “Please note that the line numbers cited refer to the revised manuscript in the Microsoft Word format”.

(3) Approximately 20-25% of SGCs are bam+, dad-LacZ+. Firstly, how do the authors explain this? Secondly, of the 70-75% of SGCs that have no/low BMP signaling, the authors should perform additional characterization using markers that are expressed in GSCs (i.e., Sex lethal and nanos).The authors did not perform additional staining for GSC-enriched protein like Sex lethal and nanos.

The 70-75% of SGCs that have low BMP signaling display the following characteristics: (1) dot-like spectrosomes, (2) positivity for *Dad*-lacZ, and (3) absence of *bamP*-GFP expression. This combination of traits is sufficient to classify them as GSC-like cells. Neither Sex lethal nor Nanos is expressed exclusively in GSCs (Chau et al., 2009; Li et al., 2009), rendering them unsuitable for distinguishing GSC-like from cystoblast-like cells.

(4) All experiments except Fig. 1I (where a single germarium with no quantification) were performed with nos-Gal4, UASp-flp. Have the authors performed any of the phenotypic characterizations (i.e., figures other than figure 1) with hs-flp?In the rebuttal, the authors stated that they used nos>flp for all figures except for Fig. 1I. It would be more convincing for them to prove in Fig. 1 than there is not phenoytpic difference between the two methods and then switch to the nos>FLP method for the rest of the paper.

We appreciate this suggestion. These data are included in Figure 1-figure supplement 3 in the revised manuscript.

(5) Does the number of SGCs change with the age of the female? The experiments were all performed in 14-day old adult females. What happens when they look at young female (like 2-day old). I assume that the nos>flp is working in larval and pupal stages and so the phenotype should be present in young females. Why did the authors choose this later age? For example, is the phenotype more robust in older females? or do you see more SGCs at later time points?The authors did not supply any data to prove that the clones were larger in 14-day-old flies than in younger flies. Additionally, the age of "younger" flies was not specified. Therefore, the authors did not satisfactorily answer my concern.

We appreciate this critical comment. Figure 1J includes the SGC phenotype data from 1-, 7-, and 14-day-old flies. Both 1- and 7-day-old flies are younger flies in our analyses. The evidence that germline clones were larger in 14-day-old flies than in younger flies was provided in Figure 1-figure supplement 2 in the revised manuscript.

(6) Can the authors distinguish one copy of GFP versus 2 copies of GFP in germ cells of the ovary? This is not possible in the Drosophila testis. I ask because this could impact on the clonal analyses diagrammed in Fig. 4A and 4G and in 6A and B. Additionally, in most of the figures, the GFP is saturated so it is not possible to discern one vs two copies of GFP.In the rebuttal, the authors stated that they cannot differential one vs two copies of GFP. They used other clone labeling methods in Fig. 4 and 6. I think that the authors should make a statement in the manuscript that they cannot distinguish one vs two copies of GFP for the record.

Thank you for this suggestion. Such statement has been added in the revised manuscript (Lines 177-178).

(7) More evidence is needed to support the claim of elevated Dpp levels in bam or bgcn mutant tumors. The current results with dpp-lacZ enhancer trap in Fig 5A,B are not convincing. First, why is the dpp-lacZ so much brighter in the mosaic analysis (A) than in the no-clone analysis (B); it is expected that the level of dpp-lacZ in cap cells should be invariant between ovaries and yet LacZ is very faint in Fig. 5B. I think that if the settings in A matched those in B, the apparent expression of dpp-lacZ in the tumor would be much lower and likely not statistically significantly. Second, they should use RNA in situ hybridization with a sensitive technique like hybridization chain reactions (HCR) - an approach that has worked well in numerous Drosophila tissues including the ovary.The HCR FISH in Fig.5 of the revised manuscript needs an explanation for how the mRNA puncta were quantified. Currently, there is no information in the methods. What is meant but relative dpp levels. I think that the authors should report in and unbiased manner "number" of dpp or gbb puncta in TFs. For the germaria, I think that they should report the number of puncta of dpp or gbb divide by the total area in square pixels counted. Additionally, the background fluorescence is noticeably much higher in bamBG/delta86 germaria, which would (falsely) increase the relative intensity of dpp and gbb in bam mutants. Although, I commend the authors for performing HCR FISH, these data are still not convincing to me.

We appreciate these critical comments. Due to variable puncta sizes and frequent clustering in TF and cap cells (see Figure 5A, C), direct quantification of puncta number was unreliable. Therefore, we quantified mean fluorescence intensity instead, as described in the revised figure legend of Figure 5 (Lines 603-604). In Author response image 1 1A, B (modified from Figure 5A, C) , magenta ovals indicate empty background fluorescence areas, which appear similar between *w1118* (wild-type control) and *bam-/-* germaria. In Author response image 1, the yellow oval outlines a neighboring germarium, not an empty area (see the DAPI channel).

**Author response image 1. sa4fig1:** *In situ*-HCR results of dpp and gbb in wild-type and bam mutant germaria. Magenta ovals indicate empty areas displaying only background fluorescence. In panel (B), the yellow oval outlines a neighboring germarium, not an empty area (see the DAPI channel below).

(8) In Fig 6, the authors report results obtained with the bamBG allele. Do they obtain similar data with another bam allele (i.e., bamdelta86)?The authors did not try any experiments with the bamdelta86 allele, despite this allele being molecularly defined, where the bamBG allele is not defined.

While we agree that repeating the experiments in Figure 6 with *bamΔ86* would be helpful, our mosaic analysis strategy for two genes on different chromosome arms is technically complex (see genotypes in Source data 1). Switching from *bamBG* to *bamΔ86* would necessitate extensive and time-consuming genetic recombination. Given that both alleles induce the SGC phenotype indistinguishably (Figure 1J), we believe that repeating these experiments with *bamΔ86* would not alter our key conclusion. We appreciate your understanding regarding this technical complexity.

**Reviewer #2 (Public review):**
In the current version, Zhang et al. have made substantial improvements to the manuscript. It is now easier to read, and the data are more solid compared with the previous version, supporting their conclusion that tumor GSCs secrete stemness factors (BMPs and Dpp) to suppress the differentiation of neighboring wild-type GSCs. This study should benefit a broad readership across developmental biology, germ cell biology, stem cell biology, and cancer biology.

Thank you for your valuable comments, and we greatly appreciate them.

However, the following suggestions may further improve the clarity and rigor of the research content:(1) Clarification of sample size (n).Each germarium can contain highly variable numbers of SGCs, sometimes reaching 50-100. When reporting "n" values, the authors are encouraged to also indicate the number of germaria analyzed. For example, in lines 126-128:"Notably, 74% of SGCs (n = 132) were GFP-negative, while the remaining 26% were GFP-positive (Figure 2B, C). This suggests that SGCs can be categorized into two distinct groups: those resembling GSCs (GSC-like) and those resembling cystoblasts (cystoblast-like)." Please clarify how many germaria were examined to obtain n = 132.

We appreciate this comment. In 14-day-old fly ovaries, each germarium that met our criterion for quantifying the SGC phenotype contains approximately 1.5 SGCs (see Figure 1K). For the specific analysis of the “132” SGCs presented in Figure 2C, we did not record the number of germaria from which they originated.

In addition, it is unclear whether the authors intend to suggest that the GFP-negative SGCs are GSC-like or cystoblast-like; this point should be clarified.

Thank you for this suggestion. We intend to suggest that the *bamP*-GFP-negative SGCs are GSC-like, which information has been added in the revised manuscript (Line 129).

(2) Improvement of Fig. 6 in situ hybridization images.The in situ hybridization images in Fig. 6 are not fully convincing. The control images, in particular, would benefit from higher resolution and enlarged views of the germarium region.

Thank you for this valuable suggestion. The enlarged views of both the control and *bam-/-* germarium regions were included in Figure 5A, C in the revised manuscript.

In panel C, abundant signals are also present outside the germarium, which may complicate interpretation and should be clarified or controlled for.

In the right panel of Figure 5C, the abundant signals noted by the reviewer originate from neighboring germaria (see the DAPI channel), not from empty areas, which would be expected to show only background fluorescence. For more details, please refer to our response to Question (7) raised by Reviewer #1.

Alternatively, the authors could strengthen the in situ analysis by using bam mutants or bam dpp / bam gbb double mutants as controls to better define signal specificity.

We appreciate this comment. Homozygous *dpp* or *gbb* mutants are lethal, precluding the generation of *dpp bam* or *gbb bam* double-mutant flies. Additionally, the GFP signal was drastically reduced during our HCR processing, preventing mosaic clone analysis.

**Reviewer #3 (Public review):**
Zhang et al. investigated how germline tumors influence the development of neighboring wild-type (WT) germline stem cells (GSC) in the Drosophila ovary. They report that germline tumors generated by differentiation-arrested mutations (bam and bgcn) inhibit the differentiation of neighboring WT GSCs by arresting them in an undifferentiated state, resulting from reduced expression of the differentiation-promoting factor Bam. They find that these tumor cells produce low levels of the niche-associated signaling molecules Dpp and Gbb, which suppress bam expression and consequently inhibit the differentiation of neighboring WT GSCs non-cell-autonomously. Based on these findings, the authors propose that germline tumors mimic the niche to suppress the differentiation of the neighboring wild-type germline stem cells.Strengths:The study uses a well-established in vivo model to address an important biological question concerning the interaction between germline tumor cells and wild-type (WT) germline stem cells in the Drosophila ovary. If the findings are substantiated, this study could provide valuable insights that are applicable to other stem cell systems.

Thank you for your valuable comments, and we greatly appreciate them.

Weaknesses:The authors have addressed some of my concerns in the revised submission. However, the data presented do not allow the authors to distinguish whether the failed differentiation of WT stem cells/germline cells results from "arrested differentiation due to the loss of the differentiation niche" or from "direct inhibition by tumor-derived expression of niche-associated molecules Dpp and Gbb".

Blocking Dpp or Gbb secretion specifically from germline tumor cells promoted differentiation of neighboring wild-type germ cells (Figure 6). This indicates that BMP ligands secreted by germline tumors are required to inhibit this differentiation. However, we cannot rule out the possibility that disruption of the differentiation niche also contributes to the SGC phenotype, a point highlighted in the manuscript (Line 204).

The critical supporting data, HCR in situ results, are not sufficiently convincing.

Below, we provide a point-by-point reply addressing each of your specific recommendations.

**Recommendations for the authors:**

**Reviewer #3 (Recommendations for the authors):**
It's a surprising that the authors failed to induce germline tumors at the adult stage, as this has been reported by many labs and would allow for time course analysis of SGC phenotype. As a result, the data in this manuscript address only events occurring after the germline tumor formation (with clonal induction at larval stage) and focus on the already presene "arrested wild-type germ cells", without providing insight into the process of by which these arrested germ cells are formed.

In our hands, inducing germline clones by the *hs-FLP* method at the adult stage was efficient in males but not in females, despite subjecting adult flies to intensive heat-shock at 37°C.

The HCR in situ data exhibit a high background.

Regarding the background issue, please see our response to Reviewer #1’s Question (7).

First, the signal appears stronger in TF cells than in cap cells.

As demonstrated by Li et al. (Li et al., 2016), *dpp*-lacZ (P4-lacZ) signals are also stronger in TF cells than in cap cells (see their Figure 4D').

Second, both dpp and gbb are detected broadly in somatic cells including escort cells. These observations are inconsistent with published data.

As shown in Figure 5A and C, *dpp* and *gbb* were detected broadly in somatic cells of *bam-/-* germaria, but not in those of *w1118* (wild-type) controls. To our knowledge, no previous study has reported the expression pattern of these ligands in a *bam* mutant background.

To demonstrate the tumor-derived dpp and gbb, the HCR in situ analysis could be performed in the germarium with mosaic clones. If these niche-associated molecules are indeed expressed in tumor cells, the authors should observe a mosaic expression pattern of these molecules, with signal "ON" in tumor cells and "OFF" in neighbouring arrested germ cells.

This is a great idea and was indeed our original approach. However, GFP signal was drastically reduced during our HCR processing, ultimately precluding mosaic clone analysis.

References

Chau, J., Kulnane, L.S., and Salz, H.K. (2009). Sex-lethal facilitates the transition from germline stem cell to committed daughter cell in the Drosophila ovary. Genetics 182, 121-132.

Li, X., Yang, F., Chen, H., Deng, B., Li, X., and Xi, R. (2016). Control of germline stem cell differentiation by Polycomb and Trithorax group genes in the niche microenvironment. Development 143, 3449-3458.

Li, Y., Minor, N.T., Park, J.K., McKearin, D.M., and Maines, J.Z. (2009). Bam and Bgcn antagonize Nanos-dependent germ-line stem cell maintenance. Proc Natl Acad Sci U S A 106, 9304-9309.